# Divergent architecture of the heterotrimeric NatC complex explains N-terminal acetylation of cognate substrates

Stephan Grunwald [1,2], Linus V. M. Hopf [1,2], Tobias Bock-Bierbaum[1], Ciara C. M. Lally[3], Christian M. T. Spahn[3] & Oliver Daumke [1,2 ✉]

The heterotrimeric NatC complex, comprising the catalytic Naa30 and the two auxiliary subunits Naa35 and Naa38, co-translationally acetylates the N-termini of numerous eukaryotic target proteins. Despite its unique subunit composition, its essential role for many aspects of cellular function and its suggested involvement in disease, structure and mechanism of NatC have remained unknown. Here, we present the crystal structure of the *Saccharomyces cerevisiae* NatC complex, which exhibits a strikingly different architecture compared to previously described N-terminal acetyltransferase (NAT) complexes. Cofactor and ligand-bound structures reveal how the first four amino acids of cognate substrates are recognized at the Naa30–Naa35 interface. A sequence-specific, ligand-induced conformational change in Naa30 enables efficient acetylation. Based on detailed structure–function studies, we suggest a catalytic mechanism and identify a ribosome-binding patch in an elongated tip region of NatC. Our study reveals how NAT machineries have divergently evolved to N-terminally acetylate specific subsets of target proteins.

[1] Department of Crystallography, Max Delbrück Center for Molecular Medicine, 13125 Berlin, Germany. [2] Institute of Chemistry and Biochemistry, Freie Universität Berlin, 14195 Berlin, Germany. [3] Institute of Medical Physics and Biophysics, Charité - Universitätsmedizin Berlin, 10117 Berlin, Germany. ✉email: oliver.daumke@mdc-berlin.de

N-terminal acetylation (Nt-acetylation) is one of the most abundant covalent protein modifications, affecting ~50% of the yeast and 80–90% of the human proteome[1–3]. The transfer of an acetyl group from acetyl-coenzyme A to the free α-amino group on the N-terminus of substrate proteins is catalyzed by N-terminal acetyltransferases (NATs), and is considered to be irreversible, as no Nt-deacetylase has been identified so far[4]. The acetyl group alters the electrostatic properties of the substrate N-terminus, which can affect protein folding, stability, half-life, interactions, and subcellular targeting[5–11]. Alteration of Nt-acetylation is implicated in several diseases, including cancers and developmental disorders[12].

All eukaryotes contain five conserved NATs, termed NatA to NatE[13], which act co-translationally on nascent polypeptide chains, as they emerge from the ribosomal exit tunnel[14–16]. In addition, higher eukaryotes express the Golgi-bound NatF[17], while plants and animals possess the kingdom-specific NatG[18] and NatH[19], respectively. All NATs contain a catalytic subunit, which belongs to the GCN5-related N-acetyltransferase (GNAT) superfamily[20]. In addition, NatA and NatB possess one, and NatC two auxiliary subunits[4]. NatE is composed of the catalytic subunit Naa50 and both NatA subunits: the catalytic subunit Naa10 and the auxiliary subunit Naa15 (refs. [14,21,22]).

NatA has the largest number of substrates and acetylates N-termini bearing a small amino acid at position 1, after the initiator methionine has been removed by methionine aminopeptidases (MetAP)[2,9,23,24]. NatB acetylates N-termini starting with methionine, followed by a residue with an acidic or amide side chain[25–27]. NatC acetylates protein N-termini starting with methionine, followed by a hydrophobic or amphipathic amino acid[9,25,28–33], with amino acids at positions 3 and 4 also contributing to NatC recognition[34]. The substrate specificity profiles of NatE and NatF partially overlap with that of NatC in vitro[3,21] or when expressed ectopically in yeast[35]. Together, these three Nt-acetylate ~21% of the human proteome[4].

The evolutionarily conserved heterotrimeric NatC complex is composed of the catalytic subunit Naa30, a large auxiliary subunit Naa35 and a small auxiliary subunit Naa38 (refs. [13,30,31]). NatC-dependent acetylation targets various substrate proteins to specific subcellular sites, including tRNA-specific methyltransferase Trm1-II to the inner nuclear membrane[36], the human ADP-ribosylation factor-like protein 8b to lysosomes[31,37] and Arf-related protein 1 (hARFRP1) and its yeast ortholog ADP-ribosylation factor-like protein 3 (yArl3) to the trans-Golgi network (TGN). This latter localization is mediated by an interaction of the acetylated N-terminus with the trans-Golgi-specific transmembrane protein hSys1/Sys1p (refs. [32,38]). Interestingly, while depletion of hARFRP1 in HeLa cells alters the structure of the TGN[39], depletion of hNaa30 induces the fragmentation of the entire Golgi stack[40], suggesting additional NatC targets at the Golgi.

The wide range of biological processes that include targets for NatC-mediated Nt-acetylation, make it essential for proper cell growth and development in eukaryotes. In yeast, deletion of any of the three subunits results in loss of in vivo NatC activity and diminished growth at 37 °C on media containing non-fermentable carbon sources. It has therefore been suggested that NatC is required for mitochondrial function[30]. Indeed, depletion of human Naa30 results in reduced expression of mitochondrial proteins, loss of mitochondrial membrane potential, and mitochondrial fragmentation[33]. NatC-dependent Nt-acetylation of the L-A virus major capsid protein Gag is required for viral particle assembly in Saccharomyces cerevisiae[28,30]. NatC is also crucial for zebrafish development; knockouts of NAA30 or NAA35 led to decreased cell proliferation, increased apoptosis, and poor blood vessel formation, resulting in embryonic lethality[41]. In the plant

kingdom, downstream targets of NatC Nt-acetylation include photosystem II core proteins D1 and CP47 in Arabidopsis thaliana, with implications for photosynthesis and plant growth[42]. In addition, knockdown of NatC subunits in human cells results in reduced cell growth and p53-dependent apoptosis[31].

A strong upregulation of NAA30 has been observed in glioblastoma, and a NAA30-knockdown in glioblastoma-initiating cells (GICs) reduced their viability, sphere-forming ability, and hypoxia tolerance[43]. Moreover, mice transplanted with GICs in which NAA30 was knocked down show prolonged survival compared to control animals transplanted with unmodified GICs, indicating that NatC may serve as a therapeutic target in cancer. Interestingly, a nuclear localization of Naa30 was observed in GICs and, more sporadically, in neuronal stem cell cultures[43]. The nuclear localization is specific to a splice variant of human NAA30, which encodes a truncated protein missing parts of the GNAT-fold. The truncated version is also abundantly expressed in thyroid cancer tissues and other human cancer cell lines[44]. In a recent study, a potentially pathogenic de novo mutation in NAA35 was identified in patients with cerebral palsy, a heterogeneous group of disorders affecting movement and posture[45].

Recently, crystal and cryo-EM structures of several NATs have been reported[21,24,46–52], but the structure of the heterotrimeric NatC complex has remained elusive. Here, to obtain insights into the tertiary and quaternary assembly of NatC, we determined its structure by X-ray crystallography and elucidated the mechanism of substrate recognition. A structure–function approach yielded insights into substrate specificity and catalysis, leading us to propose a refined reaction scheme for NatC catalysis. We also identified a ribosome-binding patch on the NatC surface and suggest a model for the NatC–ribosome complex.

## Results

**Overall structure of NatC.** To prepare the NatC complex for structural studies, the three subunits of S. cerevisiae NatC were co-expressed in Escherichia coli (Supplementary Figs. 1 and 2). The catalytic subunit Naa30 was designed as a truncation construct lacking 17 non-conserved residues at the C-terminus, analogously to a previously crystallized NatA construct[24]. During an initial purification, a partial proteolytic degradation of eleven non-conserved residues at the C-terminus of subunit Naa38 was identified by MALDI-MS, and, consequently, these 11 residues were also deleted (Supplementary Fig. 3a, b). The final homogeneous NatC preparation used for kinetic and structural studies (Supplementary Fig. 3c, d) therefore contained subunits Naa30ΔC17 (residues 1–159), full-length Naa35 (residues 1–733), and Naa38ΔC11 (residues 1–77).

Structures of the selenomethionine-derivatized (NatC, apo) and native, CoA-bound (NatC•CoA) complex were determined to 2.40 and 2.45 Å resolution, respectively. Both crystallized in space group $P2_12_12_1$. The derivatized structures were solved by single-wavelength anomalous diffraction (SAD) and refined to $R_{work}$ and $R_{free}$ values of 19.5% and 22.3%, respectively (Table 1). The structure of CoA-bound NatC was solved by molecular replacement using the ligand-free NatC structure and refined to $R_{work}$ and $R_{free}$ values of 20.3% and 23.6%, respectively.

NatC forms a heterotrimeric complex, containing the catalytic subunit Naa30, the small auxiliary subunit Naa38 and the large auxiliary subunit Naa35 (Fig. 1a). Naa30 adopts the typical, mixed α–β, GNAT-fold. The central conserved CoA-binding motif[20] harbors a CoA ligand in the NatC•CoA structure (Supplementary Fig. 4a). With RMSDs of 1.6 and 1.7 Å, Naa30 is most similar to the archaeal NAT ortholog Ard1 (ref. [53]) and the Naa10 subunit of the Schizosaccharomyces pombe NatA complex[24], respectively (Supplementary Fig. 4b).

**Table 1 Data collection and refinement statistics.**

|  | NatC, apo (SeMet)[a] | NatC•CoA (native)[a] | NatC•CoA•MFHLV (native)[a] | NatC•CoA•MLRFV (native)[a] |
|---|---|---|---|---|
| **Data collection** |  |  |  |  |
| Space group | $P2_1 2_1 2_1$ | $P2_1 2_1 2_1$ | $P2_1 2_1 2_1$ | $P2_1 2_1 2_1$ |
| Cell dimensions |  |  |  |  |
| $a, b, c$ (Å) | 48.12, 140.42, 166.41 | 48.04, 139.79, 166.56 | 48.30,, 139.72 166.82 | 48.14, 134.93, 165.74 |
| $\alpha, \beta, \gamma$ (°) | 90, 90, 90 | 90, 90, 90 | 90, 90, 90 | 90, 90, 90 |
| Wavelength (Å) | 0.9797 | 0.9184 | 0.9184 | 0.9184 |
| Resolution (Å) | 46.23–2.40 (2.48–2.40)[b] | 45.43–2.45 (2.54–2.45) | 45.65–2.99 (3.10–2.99) | 43.41–2.75 (2.85–2.75) |
| $R_{sym}$ (%)[c] | 12.8 (133.2) | 14.8 (159.5) | 16.6 (94.6) | 19.4 (150.8) |
| $I/\sigma (I)$[c] | 11.3 (1.2) | 10.9 (1.1) | 8.4 (1.5) | 9.3 (1.2) |
| $CC_{1/2}$ (%)[c] | 99.8 (56.2) | 99.7 (58.6) | 99.1 (56.4) | 99.6 (54.1) |
| Completeness (%)[c] | 99.6 (97.9) | 99.9 (99.4) | 98.7 (98.6) | 99.5 (96.0) |
| Redundancy[c] | 7.0 (6.4) | 6.6 (6.8) | 4.0 (4.2) | 6.6 (6.6) |
| **Refinement** |  |  |  |  |
| Resolution (Å) | 46.23–2.40 (2.48–2.40) | 45.43–2.45 (2.54–2.45) | 45.65–2.99 (3.10–2.99) | 43.41–2.75 (2.85–2.75) |
| No. of reflections[d] | 45,030 (4302) | 42,170 (4132) | 23,299 (2274) | 28,783 (2749) |
| $R_{work}/R_{free}$[e] (%) | 19.5/22.3 | 20.3/23.6 | 19.6/24.8 | 20.4/24.8 |
| No of atoms |  |  |  |  |
| Protein | 7718 | 7806 | 7753 | 7734 |
| Ligand/ion | 100 | 122 | 86 | 84 |
| Water | 181 | 190 | 52 | 93 |
| $B$ factors (Å$^2$) |  |  |  |  |
| Average | 65.7 | 64.5 | 63.5 | 67.4 |
| Protein | 66.0 | 64.6 | 63.6 | 67.6 |
| Ligand/ion | 72.5 | 79.4 | 67.4 | 78.3 |
| Water | 51.6 | 50.2 | 39.8 | 44.4 |
| RMS deviations |  |  |  |  |
| Bond lengths (Å) | 0.004 | 0.004 | 0.003 | 0.004 |
| Bond angles (°) | 0.90 | 0.92 | 0.86 | 0.85 |
| Ramachandran plot (%) |  |  |  |  |
| Most favored | 98.5 | 98.9 | 98.6 | 98.5 |
| Allowed | 1.5 | 1.1 | 1.4 | 1.5 |
| Outlier | 0 | 0 | 0 | 0 |
| Rotamer outliers (%) | 0.3 | 0.5 | 0.34 | 0.68 |
| Clashscore | 4.0 | 3.9 | 3.9 | 5.4 |
| PDB accession code | 6YGA | 6YGB | 6YGC | 6YGD |

*SeMet* selenomethionine.
[a]Data are from one crystal for each structure.
[b]Values in parentheses are for the highest resolution shell.
[c]Values are calculated for separately counted Friedel's pairs in the SeMet-substituted apo structure.
[d]Friedel pairs were merged during refinement.
[e]$R_{free}$ was calculated with 5% of the reflection data.

Naa35 is mostly α-helical and, in addition, contains three short β-strands in the N-terminal region and two β-strands at the C-terminus (Supplementary Fig. 2). Helix α21, together with the C-terminal end of α20, protrude by ~30 Å from the central body, thereby forming an extension, which we refer to as the "Naa35 tip". A DALI search did not yield any closely related relatives of Naa35, indicating a unique fold.

In agreement with previous predictions[54], Naa38 adopts an Sm fold, with an N-terminal α-helix, followed by a strongly bent five-stranded β-sheet[55]. An additional short α-helix is present at the C-terminus. A DALI search revealed that Naa38 is most similar to the spliceosomal Lsm4 (RMSD of 1.8 Å over 72 Cα atoms; Supplementary Fig. 4c–f), which is part of the donut-shaped, hetero-heptameric Lsm2–8 ring that binds snRNA U6 in its center[56]. However, Naa38 differs from other Lsm proteins, as it lacks the conserved IRG motif that mediates association with small nuclear RNAs[54].

**Naa35 is the central assembly hub of the NatC complex.** The large auxiliary subunit Naa35 acts as the central assembly hub of

the NatC complex, forming extensive interactions with Naa30 and Naa38 (Fig. 1b). The elongated Naa35 N-terminus wraps around almost the entire circumference of Naa38, burying a surface area of 1970 Å$^2$. As Naa35 extends around Naa38, it connects it to Naa30, thereby limiting the direct contact between the catalytic and the small auxiliary subunit to 260 Å$^2$. The remainder of Naa35 wraps around Naa30 in a more condensed, ring-like structure covering three quarters of its circumference and burying another 1890 Å$^2$.

The quaternary assembly of the three NatC subunits results in the formation of a tunnel in the center of the NatC complex (Fig. 1c). This tunnel is surrounded by loop regions of Naa30 and Naa35, which comprise the peptide-binding site (see below). The tunnel extends to a deep groove that contains the active site of the complex and accommodates the acetyl-CoA cofactor-binding site.

Interactions between the three NatC subunits are mediated by several evolutionary conserved contacts (Supplementary Fig. 5). While the interface between Naa30 and Naa35 is dominated by hydrogen bonding interactions, the β1–α2–loop–β2 segment of Naa35 forms extensive hydrophobic interactions with Naa38.

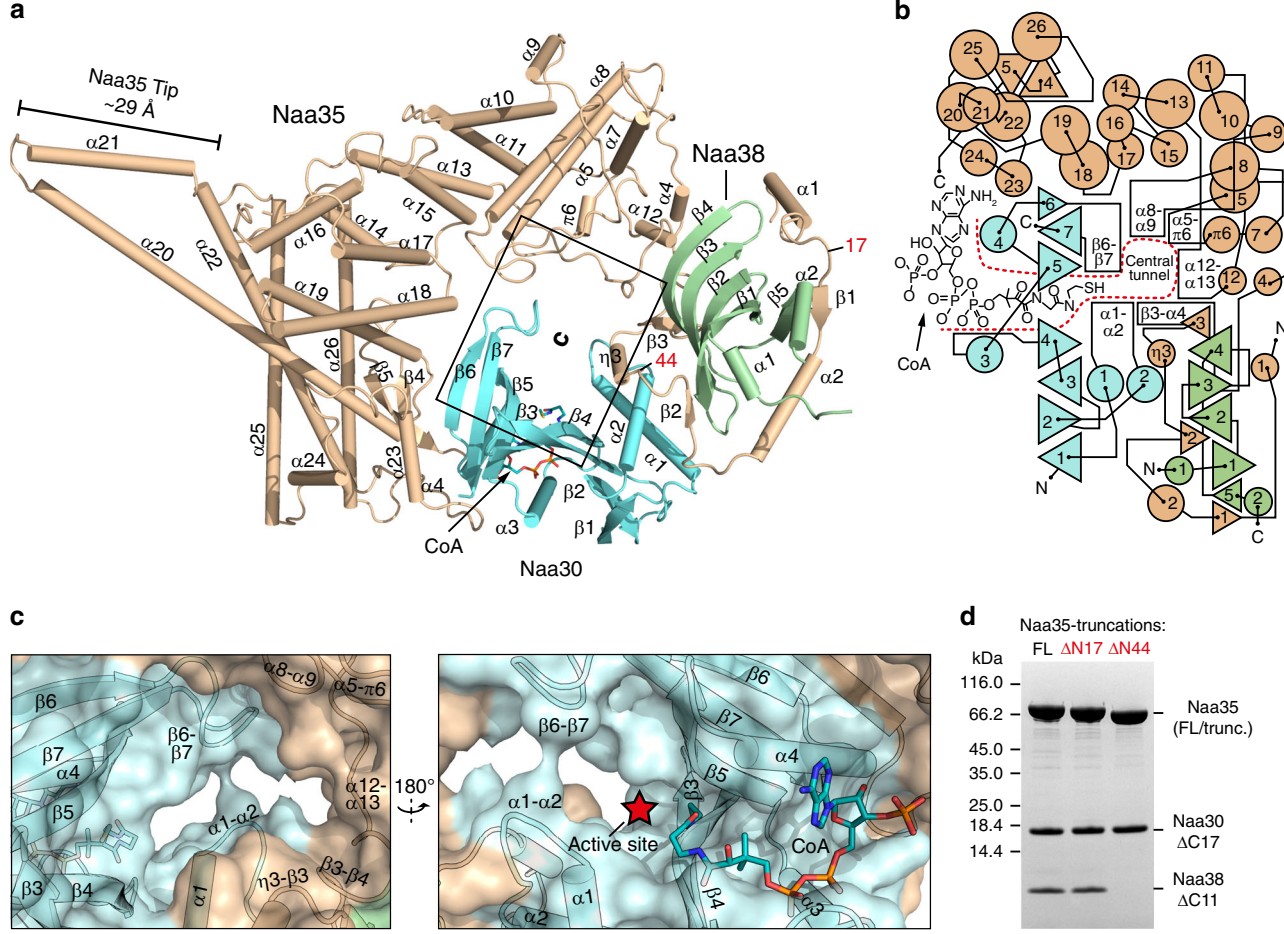

**Fig. 1 Overall structure of the NatC complex bound to coenzyme A. a** Cartoon representation of NatC bound to CoA (stick format). **b** Topology plot of NatC. Triangles represent β-strands, small triangles indicating sheets with ≤6 residues. Circles represent α-, $3_{10}$- (η), or pi- (π) helices with ≤5 residues (small circles), 6–19 residues (medium circles) or ≥20 residues (large circles). **c** Magnified view of the NatC active site (cartoon with surface representation). **d** SDS–PAGE analysis from NatC complexes affinity-purified via the Naa35 subunit, containing full-length (FL) Naa35 (residues 1–733), Naa35ΔN17 (residues 18–733), or Naa35ΔN44 (residues 44–733).

Moreover, the β-sheet of Naa38 is extended by three short β-strands from the Naa35 N-terminus, forming a bifurcated, antiparallel β-sheet. Thus, within the NatC complex, the Naa38 β-sheet is not available for homo- or hetero-oligomeric interactions, unlike in the Lsm2–8 assembly (Supplementary Fig. 4e, f). In agreement with the structural data, deletion of the first 44 residues of Naa35 disrupted the interaction with Naa38. In contrast, deletion of the non-conserved N-terminal α1 helix (Naa35ΔN17) was not sufficient to disrupt NatC integrity (Fig. 1d).

**Comparison of NatC with heterodimeric NatA and NatB complexes.** A comparison of the heterotrimeric NatC complex with the heterodimeric NatA[24] and NatB[50] complexes revealed remarkable differences in their tertiary and quaternary structures (Fig. 2). All NAT complexes contain a catalytic subunit, with a conserved GNAT architecture. While NatA and NatB have related, single auxiliary subunits, NatC possesses two unrelated auxiliary subunits. Strikingly, the relative position of the NatC catalytic and auxiliary subunits is opposite to their arrangement in NatA and NatB. Whereas the auxiliary subunits Naa15 (NatA) and Naa25 (NatB) primarily engulf the N-terminal part of their catalytic subunits, Naa35 (NatC) additionally encloses the C-terminal half. In NatC, the β6–β7 loop, which is necessary for substrate binding, is in direct contact with the auxiliary subunit

Naa35. In NatA and NatB, the β6–β7 loop makes no contact with the corresponding auxiliary subunit. Furthermore, NatA and NatB auxiliary subunits are necessary for the proper positioning of the catalytic α1–loop–α2 region, and hence for full catalytic activity[24,50]. In NatC, helix α2 of Naa30 is in contact with both auxiliary subunits Naa35 and Naa38. However, in contrast to NatA and NatB, α1 in Naa30 forms no contacts to either of the two auxiliary subunits.

**Substrate recognition in NatC.** Colorimetric activity assays were performed to kinetically characterize the activity of NatC. Short decameric peptides were used as substrates, in which the five initial N-terminal residues corresponded to the sequence of cognate NatC substrates (Fig. 3a, Table 2, and Supplementary Figs. 6 and 7). A peptide containing the N-terminus of S. cerevisiae Arl3 (yArl3) was Nt-acetylated by NatC with a $k_{cat}$ of $4.2 \pm 0.3 \, s^{-1}$ and a $K_m$ of $140 \pm 20 \, \mu M$. A comparable activity was observed for a peptide with the N-terminal sequence of hARFRP1, the human ortholog of yArl3, suggesting that species-specific differences in the N-terminal sequences between yArl3 and hARFRP1 are not due to adaption to their respective NatC ortholog. A peptide comprising the five initial N-terminal residues of the major capsid protein Gag, exhibited a $k_{cat}$ of $12 \pm 1 \, s^{-1}$ and a $K_m < 20 \, \mu M$. Therefore, the specificity

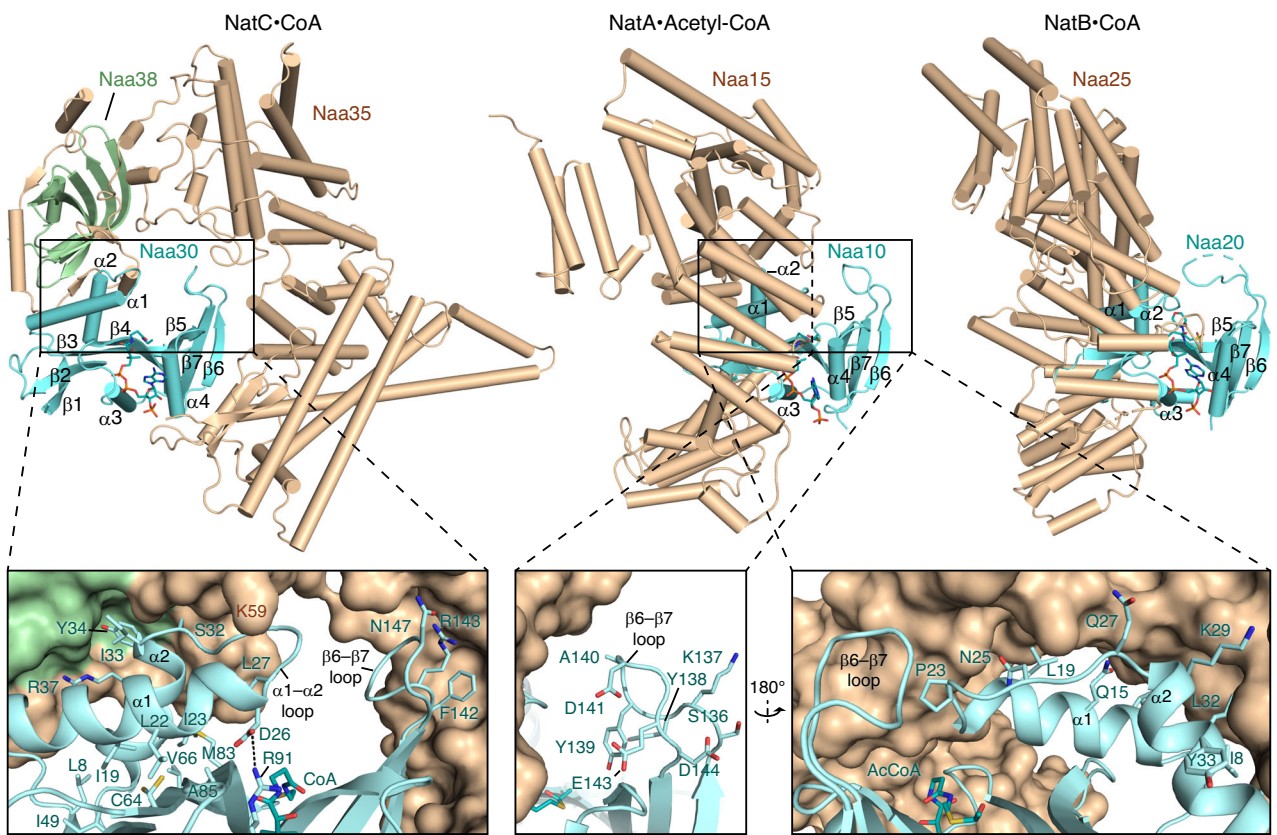

**Fig. 2 Structural comparison of the NatC complex.** Comparison of CoA-bound *S. cerevisiae* NatC with acetyl-CoA-bound *S. pombe* NatA (pdb code 4KVO) and CoA-bound *Candida albicans* NatB (5K04). The catalytic subunits (cyan) are in the same orientation. The boxed areas are magnified at the bottom and show the interactions of the α1–loop–α2 segment and β6–β7 loop (cartoon representation), with the auxiliary subunits (surface representation).

constant of NatC toward the Gag peptide is at least 20-fold higher compared to yArl3.

To obtain insights into the substrate recognition mode and the marked difference of NatC catalytic efficiencies toward the two peptides, crystal structures of NatC in complex with CoA, and either yArl3 or Gag peptide were determined to a maximal resolution of 2.99 and 2.75 Å, respectively (Table 1). The electron density of the peptide ligands was sufficient to model the first four and six amino acids of the yArl3 and Gag peptide, respectively (Fig. 3b and Supplementary Fig. 8). Moreover, clear electron density was visible for a tightly coordinated water molecule in the active site of the NatC complex.

The yArl3 and the Gag peptide formed several side chain and backbone interactions, mostly with Naa30, but also with Naa35. In both structures, the methionine sulfur atom at peptide position 1 is within hydrogen bond distance to Ser28 (Fig. 3c, d). No NatC activity was measured for a known NatA substrate (yeast threonyl-tRNA synthetase, yThrRs), which lacks an N-terminal methionine[24].

The Phe2 and Leu2 side chains of the yArl3 and Gag peptide, respectively, were positioned in a hydrophobic pocket (the "main peptide pocket"), which is formed by conserved residues from α2 and β4 of Naa30, as well as the short $3_{10}$-helix η3 from Naa35 (Fig. 3c and Supplementary Fig. 9). While F2W and F2A substitutions led to comparable $k_{cat}$ and $K_m$ values as seen for the native yArl3 peptide, the F2R, F2Y, F2L, and F2K substitutions resulted in 3–8-fold and the F2E substitution in a 40-fold increased $K_m$ value (Table 2 and Fig. 3a). Also, the L2E substitution in the Gag peptide led to an at least 13-fold decrease of $K_m$. Substitutions at position 2 to positively charged amino

acids (F2R and F2K) in yArl3 led to strong decreases in turnover numbers to 3–8%.

At peptide position 3, Gag-Arg3 forms a hydrogen bond with the protein backbone and a cation–π interaction with Naa30-Tyr144. In addition, Naa30-Glu29 interacts with the Gag peptide backbone. In the yArl3-bound structure, in contrast, Naa30-Glu29 forms an alternative hydrogen bond with yArl3-His3 and does not contact the peptide backbone. Glutamate substitutions at position 3 massively increased $K_m$ and reduced $k_{cat}$ for both peptides. Interestingly, the H3A substitution in yArl3 led to $k_{cat}$ and $K_m$ values that were similar to those of the Gag peptide, suggesting that an imidazole side chain at peptide position 3 interferes with full catalytic activity.

Only in the Gag peptide, the fourth peptide residue, Phe4, deeply inserts into a second hydrophobic surface generated by Naa30 and Naa35, the "extended pocket" (Fig. 3c and Supplementary Fig. 9). The yArl3-Leu4 side chain is at the periphery of this pocket. Glutamate substitutions at position 4 led to strong increases in $K_m$ for both peptides. For the Gag peptide, $k_{cat}$ was strongly reduced in addition. Similar to the yArl3-H3A substitution, also the yArl3-L4A variant exhibited a strong increase in $k_{cat}$ and a strongly reduced $K_m$ value. Furthermore, a peptide derived from the N-terminus of yeast actin, a known NatB substrate[50] with acidic residues at positions two and four, was not acetylated by NatC.

Val5 in the Gag peptide occupies the same position as Leu4 of yArl3, whereas residue 5 is not resolved for yArl3. Glutamate substitutions at position 5 only moderately affected the specificity constant for both peptides, indicating a minor effect of this position for binding.

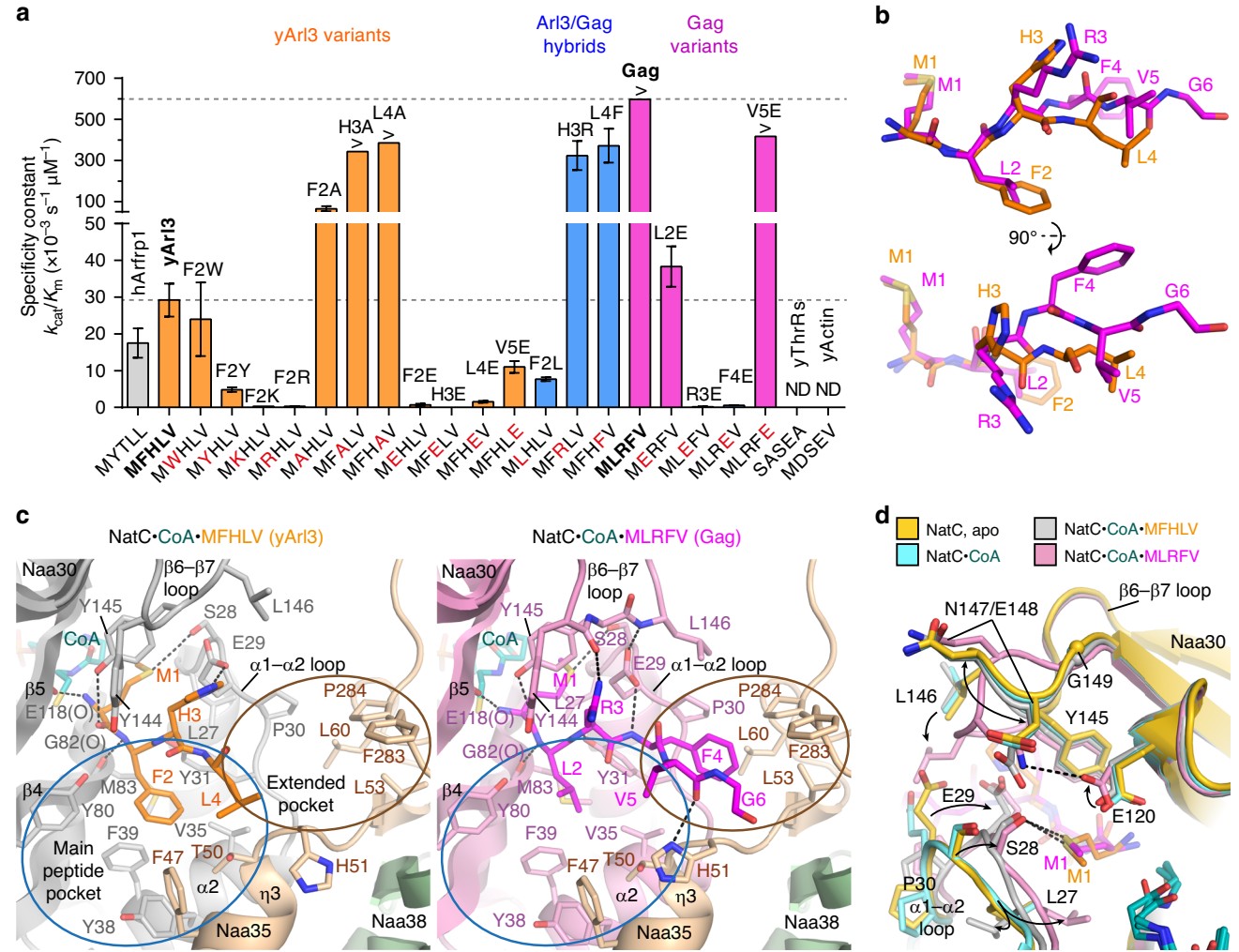

**Fig. 3 NatC substrate recognition. a** Specificity constants of NatC WT toward various decameric substrate peptides. The N-terminal five residues are shown, with all peptides ending on -GSRRR. Data represent means ± SD calculated from $k_{cat}$ and $K_m$ values of independent experiments considering the propagation of uncertainty. The number of repetitions is provided in Table 2. Four peptides exhibited $K_m$ values below the detection limit of the colorimetric assay and were assumed to be <20 μM (Supplementary Fig. 6). Thus, the bar graphs for these peptides (marked with ">") represent the lower bound of the specificity constant and no error bar is provided. **b** Substrate peptides in the NatC•CoA•MFHLV (orange) and NatC•CoA•MLRFV (magenta) structures. **c** Magnified views into the active sites of the two peptide-bound NatC complexes, shown in the same orientation. Hydrogen bonds are indicated as black dashed lines. **d** Superposition of all four NatC structures focusing on structural changes in the α1–α2 and β6–β7 loop in Naa30.

To further explore the differences between the yArl3 and Gag peptides, yArl3/Gag hybrids with single amino acid substitutions at positions 2–4 were introduced in the yArl3 peptide. While the F2L hybrid exhibited a strongly reduced specificity constant compared to yArl3, the H3R and L4F hybrids approached $K_m$ and $k_{cat}$ values of the Gag peptide (Fig. 3a and Table 2). These results highlight the importance of positions 3 and 4 for efficient substrate acetylation.

**Peptide ligand-induced conformational changes.** The higher turnover number of the Gag peptide compared to yArl3 suggests favorable structural rearrangements of active site residues. With an RMSD of 0.218 Å, binding of CoA induced only minor structural changes in the catalytic subunit compared to the apo structure. However, binding of yArl3, and, especially, the Gag substrate, led to large conformational changes (Fig. 3d and Supplementary Fig. 10a). In particular, binding of the Gag peptide induced a marked constriction of the central tunnel in NatC (Supplementary Fig. 10b). The α1–α2 loop of Naa30 moved ~2.8

or 3.5 Å upon binding of the yArl3 or Gag peptide, respectively. In the peptide-free NatC structures, Leu27 in the α1–α2 loop is positioned in a pocket in between helices α1 and α2. Binding of Gag induces a flipping of Leu27, allowing recognition of the N-terminal methionine, which is sandwiched between Leu27 and Tyr145 of Naa30. In contrast, the Leu27 side chain appears to be flexible in the yArl3 structure (Supplementary Fig. 8b).

In contrast to the yArl3 peptide, binding of the viral Gag peptide induced a large conformational rearrangement of the β6–β7 loop in Naa30. Naa30-Asn147 forms a hydrogen bond with Naa35-Gln106 in the apo and CoA/yArl3-bound NatC structures, but traversed a distance of 8.8 Å in the Gag-bound structure to form an intramolecular hydrogen bond with Naa30-Glu120. Moreover, Naa30-Leu146 moved ~5 Å toward Gag-Phe4, which complements the extended peptide pocket. Thus, NatC exhibits a sequence-specific, peptide ligand-induced conformational change of the β6–β7 loop.

Interestingly, the Naa30 β6–β7 loop in the Gag-bound structure adopts a conformation that strongly resembles that of Naa10 in the NatA complex (Supplementary Fig. 10c). This

**Table 2 Catalytic parameters for wild type and mutant NatC with various substrates.**

| Enzyme | Substrate N-terminus[a] | Substrate description | $k_{cat}$ (s$^{-1}$) | $k_{cat}$/WT[b,d] | $K_m$[c] (µM) | $K_m$/WT[b,d] | $n$ |
|---|---|---|---|---|---|---|---|
| NatC WT[b] | **MFHLV-** | **yArl3** | **4.2 ± 0.3** | **1.0** | **140 ± 20** | **1.0** | 5 |
| | MYTLL- | hARFRP1 | 3.2 ± 0.3 | 0.77 | 190 ± 40 | 1.3 | 4 |
| | MWHLV- | yArl3-F2W | 3.4 ± 0.4 | 0.80 | 140 ± 60 | 1.0 | 4 |
| | MYHLV- | yArl3-F2Y | 2.5 ± 0.1 | 0.60 | 520 ± 60 | 3.6 | 4 |
| | MKHLV- | yArl3-F2K | 0.35 ± 0.01 | 0.083 | 1190 ± 60 | 8.3 | 3 |
| | MRHLV- | yArl3-F2R | 0.135 ± 0.003 | 0.032 | 450 ± 80 | 3.1 | 3 |
| | MAHLV- | yArl3-F2A | 5.1 ± 0.3 | 1.2 | 80 ± 10 | 0.55 | 3 |
| | MFALV- | yArl3-H3A | 6.9 ± 0.5 | 1.6 | <20* | <0.14 | 3 |
| | MFHAV- | yArl3-L4A | 8 ± 1 | 1.8 | <20* | <0.14 | 4 |
| | MEHLV- | yArl3-F2E | 4 ± 1 | 0.95 | 6000 ± 2000 | 40 | 5 |
| | MFELV- | yArl3-H3E | 0.18 ± 0.01 | 0.043 | 3600 ± 200 | 25 | 3 |
| | MFHEV- | yArl3-L4E | 3.2 ± 0.1 | 0.76 | 2100 ± 300 | 15 | 3 |
| | MFHLE- | yArl3-V5E | 6.7 ± 0.3 | 1.6 | 620 ± 90 | 4.3 | 3 |
| | MLHLV- | yArl3/Gag-F2L | 5.6 ± 0.2 | 1.3 | 730 ± 50 | 5.1 | 3 |
| | MFRLV- | yArl3/Gag-H3R | 9 ± 2 | 2.1 | 27 ± 2 | 0.19 | 4 |
| | MFHFV- | yArl3/Gag-L4F | 10 ± 2 | 2.5 | 28 ± 4 | 0.19 | 4 |
| | **MLRFV-** | **Gag** | **12 ± 1** | **1.0** | **<20*** | **1.0** | **3** |
| | MERFV- | Gag-L2E | 10.3 ± 0.8 | 0.86 | 270 ± 30 | 13 | 3 |
| | MLEFV- | Gag-R3E | 0.72 ± 0.08 | 0.060 | 3800 ± 300 | 190 | 3 |
| | MLREV- | Gag-F4E | 1.01 ± 0.07 | 0.084 | 1700 ± 100 | 86 | 3 |
| | MLRFE- | Gag-V5E | 8.3 ± 0.8 | 0.70 | <20* | 1.0 | 5 |
| | SASEA- | yThrRs | ND | | | | |
| | MDSEV- | yActin | ND | | | | |
| NatC-Naa30 mutations | | | | | | | |
| L27A | MFHLV- | yArl3 | 0.010 ± 0.005 | 0.0023 | 80 ± 30 | 0.56 | 4 |
| S28A | | | 1.8 ± 0.1 | 0.43 | 270 ± 30 | 1.9 | 3 |
| E29A | | | 0.24 ± 0.01 | 0.058 | 160 ± 30 | 1.1 | 4 |
| E29Q | | | 0.18 ± 0.02 | 0.043 | 250 ± 40 | 1.7 | 3 |
| Y31F | | | 0.45 ± 0.01 | 0.11 | 230 ± 50 | 1.6 | 3 |
| Y80A | | | 2.5 ± 0.3 | 0.59 | 550 ± 70 | 3.9 | 3 |
| Y80F | | | 1.68 ± 0.01 | 0.40 | 430 ± 60 | 3.0 | 3 |
| E118A | | | 0.40 ± 0.03 | 0.10 | 320 ± 30 | 2.3 | 3 |
| E118Q | | | 0.60 ± 0.02 | 0.14 | 430 ± 60 | 3.0 | 4 |
| E120A | | | 0.88 ± 0.04 | 0.21 | 260 ± 10 | 1.8 | 3 |
| E120Q | | | 1.01 ± 0.05 | 0.24 | 300 ± 60 | 2.1 | 4 |
| Y130A | | | 0.057 ± 0.002 | 0.013 | 200 ± 60 | 1.4 | 4 |
| Y130F | | | 0.08 ± 0.01 | 0.020 | 120 ± 20 | 0.80 | 4 |
| Y145F | | | 1.02 ± 0.09 | 0.24 | 230 ± 10 | 1.6 | 3 |
| NatC-Naa35 mutations | | | | | | | |
| F47A | MFHLV- | yArl3 | 2.3 ± 0.1 | 0.55 | 160 ± 10 | 1.1 | 3 |
| K59A | | | 2.2 ± 0.3 | 0.53 | 191 ± 8 | 1.3 | 3 |
| Tip1[e] | | | 4.2 ± 0.4 | 1.0 | 180 ± 10 | 1.2 | 3 |
| NatC-Naa38 deletion[f] | MFHLV- | yArl3 | 0.27 ± 0.04 | 0.063 | 110 ± 10 | 0.75 | 3 |

*WT* wild type.
[a]N-terminal five residues of the decameric substrate peptides ending on -GSRRR. Peptides with N-terminal sequences of cognate yeast NatC substrates (yArl3 and Gag) are printed in bold letters. Single amino acid substitutions in related yArl3- and Gag-peptide variants are underlined.
[b]WT NatC construct (Naa30ΔC17, full-length Naa35 and Naa38ΔC11).
[c]$K_m$ values are for the substrates in the substrate column. The acetyl-CoA $K_m$ was calculated for NatC WT using the yArl3 substrate ($K_m$ = 30 ± 6 µM for $n$ = 4 independent experiments). Four $K_m$ values (*) were below the detection limit of the colorimetric assay and were assumed to be <20 µM (see Supplementary Fig. 6).
[d]All normalizations are relative to NatC WT with the yArl3 peptide, except for the Gag-L2E, -R3E, -F4E, and -V5E peptides, which were compared against Gag.
[e]The Naa35-Tip1 mutant contains point mutations: K500A, K501A, K503A, and K504A.
[f]The Naa38 deletion construct consist of Naa30ΔC17 and full-length Naa35 only. Where $k_{cat}$ is ND (not determined), activity could not be detected. Values for $k_{cat}$ and $K_m$ represent means ± SD of n independent experiments (see last table column).

Naa10 conformation is similar in the presence and absence of peptide substrate. As a result, the β6–β7 loop of NatA appears already primed for efficient catalysis in the peptide-free state.

**Catalytic mechanism of the NatC complex.** A structure-based mutagenesis approach was performed to characterize the catalytic mechanism of NatC. All mutations of residues close to the active site of the Naa30 subunit led to a reduction of $k_{cat}$ of at least 40% (Table 2 and Fig. 4a, b). Reductions to ≤10% of wild-type (WT) activity were observed for residues Glu29, Glu118, and Tyr130, indicating their crucial role for catalysis. The L27A mutant showed the most severe reduction in $k_{cat}$ with a minor impact on

$K_m$, suggesting that affinity toward the substrate was not negatively affected.

Mutations in the large auxiliary subunit Naa35, K59A (Supplementary Fig. 10a), and F47A (Fig. 3c) had only moderate effects on catalytic parameters (Table 2). A NatC construct expressed and purified without its small auxiliary subunit Naa38 (ΔNaa38) showed a strong decrease in $k_{cat}$ to 6% of WT activity, while the $K_m$ remained almost identical to WT, indicating that the small subunit Naa38 is crucial for NatC activity. On the basis of this mutagenesis study and previous functional data of other NAT complexes, we propose a catalytic mechanism for the NatC complex (Fig. 4c), which is detailed in the discussion.

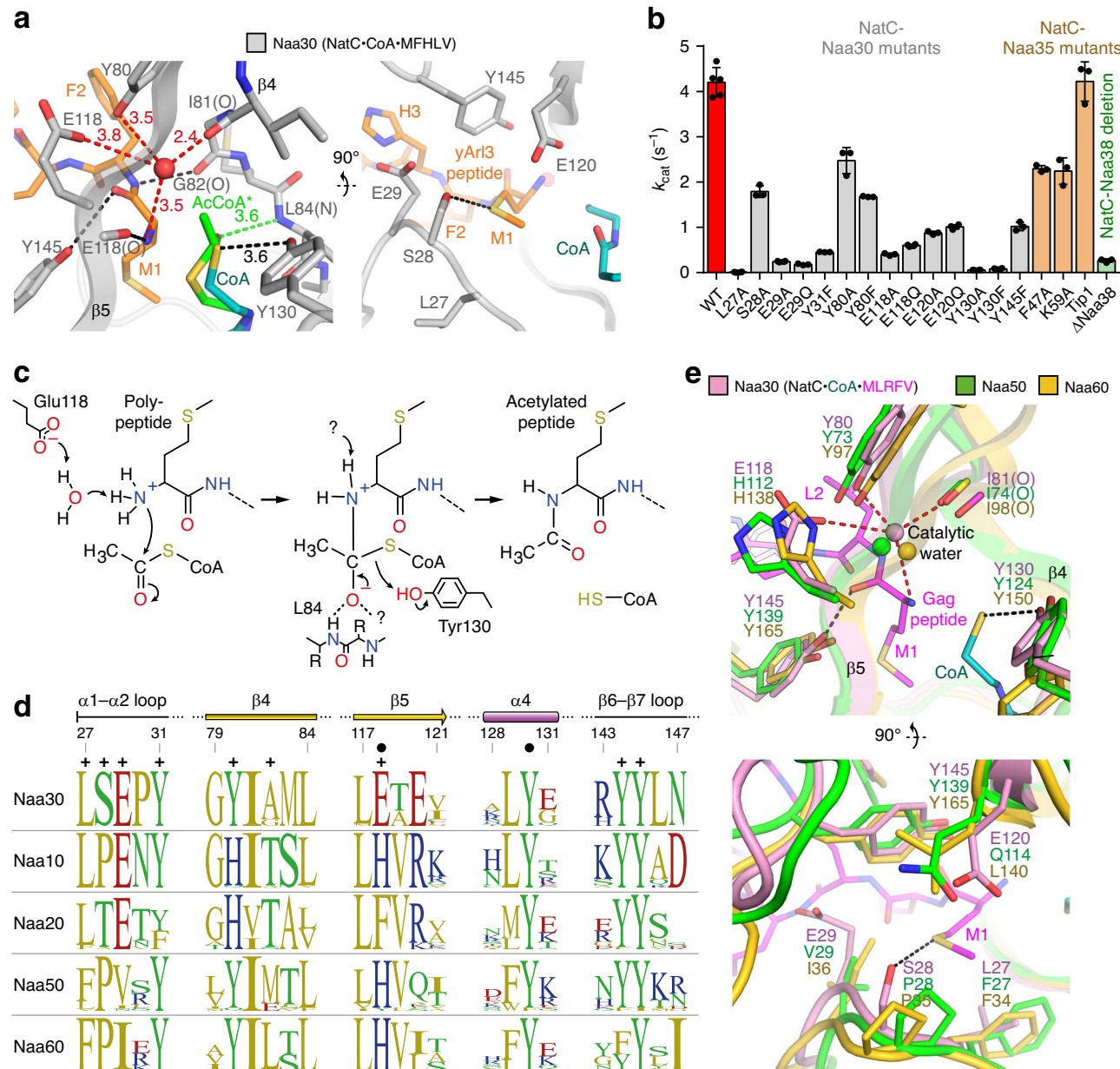

**Fig. 4 Proposed catalytic mechanism for the NatC complex. a** The active site of the NatC complex in NatC•CoA•MFHLV. The proposed catalytic water is indicated by a red sphere. The position of acetyl-CoA (green) was modeled based on a superposition of the Ard1•Acetyl-CoA structure (4LX9) with Naa30. **b** Turnover numbers ($k_{cat}$) of NatC WT and NatC mutants with the yArl3 substrate peptide. Data represent mean ± SD of independent experiments. The number of repetitions is provided in Table 2. **c** Proposed catalytic mechanism for the NatC complex (see discussion). **d** Sequence logos of regions near the active site, generated from multiple sequence alignments of 18 different species (see "Methods"). Black circles indicate proposed catalytic residues and + signs, substrate-binding residues. **e** Superposition of NatC•CoA•MLRFV, Naa50•CoA-Ac-MGLP (3TFY) and Naa60•CoA-Ac-MKAV (5ICV). For simplicity, only NatC ligands are shown. The proposed catalytic water (sphere) is indicated in all NATs.

**NatC–ribosome interaction.** All major NATs, i.e., NatA to NatE, are associated with mono- and polyribosomes[14–16]. Using co-sedimentation assays, two conserved electropositive regions (EPR) in NatA were shown to mediate its ribosome interaction[57]. We also identified several distinct EPRs in NatC (Fig. 5a). The four most prominent ones, EPR1–EPR4, were selected for mutational studies. EPR2, located on the Naa35 tip, contains 11 positively charged residues, which were divided into two sets (Tip1 and Tip2). For each NatC mutant, between three and four positively charged residues were substituted with alanine. Our co-sedimentation assay showed that the electropositive Naa35 tip region is necessary for association of NatC with the ribosome,

whereas ribosome binding was not affected for the other EPR mutants (Fig. 5b, c and Supplementary Fig. 11). The Tip1 mutant showed no significant difference in $k_{cat}$ and $K_m$ compared to NatC, arguing against major structural disturbances in NatC induced by the mutations. Interestingly, Naa38 does not seem to be required for the NatC–ribosome association, as the Naa35–Naa30 heterodimer associated with ribosomes similarly to NatC.

*S. cerevisiae NAA35*-deletion strains (*naa35Δ*) were reported to exhibit reduced growth on non-fermentable carbon sources[30,58,59], and we confirmed these results (Fig. 5d and Supplementary Fig. 12). In complementation experiments, both

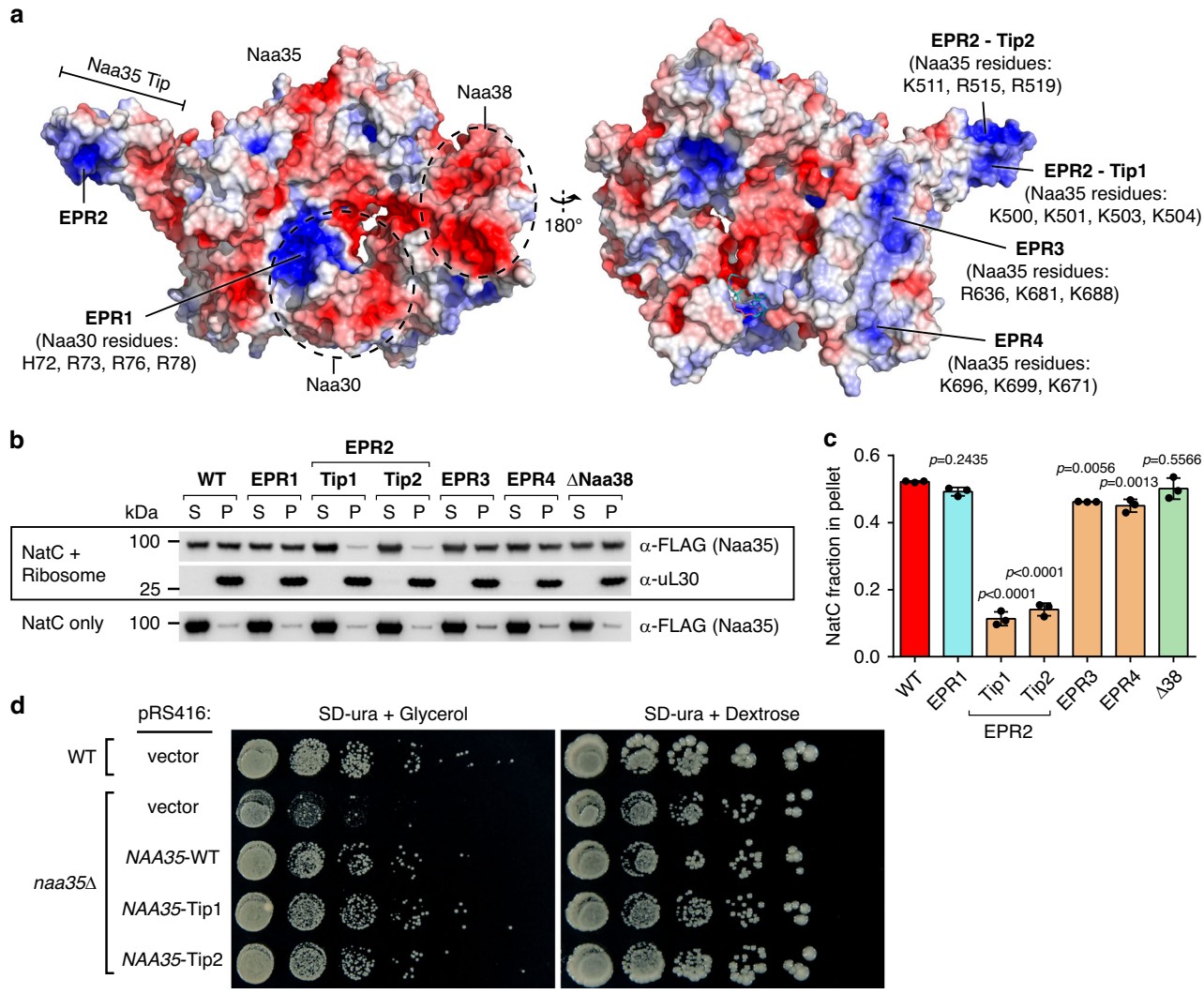

**Fig. 5 NatC–ribosome interaction. a** Electrostatic surface potential projected on the surface of the NatC•CoA structure. Residues within electropositive regions (EPRs) that were substituted with alanine in NatC mutants constructs are specified in brackets. **b** Representative western blot of a NatC/ribosome co-sedimentation assay. NatC constructs in the supernatant (S) and pellet (P) fractions were immunodetected via a FLAG-tag at the Naa35 N-terminus, yeast ribosomes via the ribosomal protein uL30. Co-sedimentations assays were performed in triplicate (all replicates are shown in Supplementary Fig. 11). **c** Quantification (chemoluminescence band intensities) of the NatC fraction in the pellet (P/(S + P)). Data represent mean ± SD ($n = 3$ independent experiments). NatC mutants were compared against NatC WT using a one-way ANOVA analysis with Dunnetts correction. **d** Serial tenfold dilutions of *S. cerevisiae* WT (BY4741) and *naa35Δ* (Y00294) strains, transformed with a pRS416 yeast centromere vector, carrying no insert (vector), *NAA35*-WT or the *NAA35*-Tip1 or *NAA35*-Tip2 mutant genes. Cells were grown at 37 °C for 5 days on SD-ura agar plates, supplemented with 3% glycerol or 2% dextrose, respectively.

*NAA35*-WT, and the Tip1 or Tip2 variants rescued the growth phenotype of the *naa35Δ* strain on glycerol. Thus, the EPR2 region in the Naa35 tip region appears not essential for yeast growth on glycerol.

## Discussion

In this work, we report the crystal structure of the conserved heterotrimeric NatC complex. We show that the large auxiliary subunit Naa35 is the central scaffold of NatC, which forms extensive interactions with Naa30 and Naa38. The architecture of the complex is strikingly different from that of other NAT complexes, which is mainly due to the unique architecture and interaction network of the Naa35 subunit. By characterizing the peptide-binding site located in between the catalytic Naa30 and the Naa35 subunits, our works explains the substrate specificity of NatC. Furthermore, we reveal how substrate binding

leads to correct positioning of the catalytic machinery, and finally identify residues important for catalysis and ribosome interaction.

Naa35 mediates the unique assembly of NatC into a highly intertwined complex. Its N-terminus mediates the interaction with Naa38, while the C-terminal residues engage in an intricate interaction network with the catalytic subunit. Besides this scaffolding function, Naa35 also mediates the interaction with the ribosome via an elongated tip region. We did not find close relatives of Naa35 in the PDB database, suggesting that Naa35 has evolved a specialized function in the NatC complex. Deletion of any of the three NatC subunits in yeast leads to a complete loss of Nt-acetylation of a NatC model peptide[30]. On the other hand, Nt-acetylation of yArl3 is abolished in a *naa30Δ* strain, but still present in *naa38Δ* yeast cells[38]. Accordingly, we show that a Naa38-deletion construct of NatC still displayed residual activity

for a yArl3 peptide. The influence of Naa38 on catalytic activity is likely indirect, as Naa38 is far away from the catalytic subunit and the two subunits share only a small interaction interface. Instead, Naa38 seems to stabilize the Naa35 N-terminus, which runs in between Naa30 and Naa38, and forms the distal end of the extended peptide-binding pocket. When expressed on its own, yeast Naa30 aggregated in our hands, highlighting its dependence on Naa35. However, the *Arabidopsis* homolog of Naa30 can functionally replace all three NatC subunits in yeast, and *AtNaa35* knockout plants show no obvious phenotype[42]. Thus, in contrast to the yeast counterpart, plant Naa30 can either act alone or as part of a different multiprotein complex.

NatC acetylates substrates with an initial N-terminal methionine followed by a hydrophobic/amphipathic amino acid[9,25,29–34]. This substrate preference can well be explained by our two substrate-bound NatC structures. In the peptide-free NatC structures, Leu27 is positioned in a hydrophobic pocket in between helices α1 and α2. In the Gag-bound structure, we observed a repositioning of Leu27 from a hydrophobic pocket within Naa30 toward the N-terminal methionine of the substrate peptide, whereas Leu27 appeared flexible in the yArl3-bound structure. We envision that the repositioning of Leu27 upon substrate-binding primes the catalytic machinery, as the L27A mutant exhibited the strongest catalytic reduction in our assay. The substrate's second amino acid is placed in the hydrophobic main peptide pocket and shows a preference for phenylalanine, tryptophane, or alanine. The latter is unlikely to exist in vivo, as MetAP usually cleaves the amino-terminal methionine when it precedes a small amino acid[60], unless subsequent inhibitory residues (e.g., Pro at position 3) restrain MetAP activity[61]. Terecero et al. have already qualitatively shown in yeast that also substrate positions 3 and 4 are important for acetylation by NatC, as glutamate substitutions at these positions prevented Nt-acetylation[34]. Our peptide-bound structures reveal that Gag-Arg3 interacts with Tyr144 of the β6–β7. Furthermore, a contact of Naa30-Glu29 with the Gag peptide backbone stabilizes active site loops α1–α2 and β6–β7 (Fig. 3c). For the yArl3 peptide, this interaction is sterically prevented by a hydrogen bond of yArl3-His3 with Naa30-Glu29, which may partially account for the reduced specificity constant of yArl3 compared to Gag. When the steric restraints are released by the H3A substitution and, possibly, the L4A substitution in yArl3, efficient catalysis is restored. Furthermore, the Gag peptides explores an extended peptide pocket, which can accommodate another large hydrophobic amino acid at peptide position four. Correspondingly, the yArl3-L4F substitution leads to reduction in $K_m$. Our data thus suggest that amino acids 3 and 4 of the Gag peptide induce sequence-specific structural rearrangements in the active site that are favorable for catalysis. This can explain the high specificity constant of the Gag peptide, which likely reflects an evolutionary adaption of the L-A virus to ensure that all Gag protomers are Nt-acetylated to allow an error-free, seamless assembly of the viral capsid. Interestingly, three mitochondrial proteins acetylated by NatC in yeast share the same four N-terminal amino acids as Gag[34], and may therefore be modified in a similarly efficient manner.

Even though NatC, NatE, and NatF share substantially overlapping substrate specificity in vitro[9], they may have a smaller overlap in vivo. In yeast, the Naa50 subunit of the NatE complex lacks the optimal acetyl-CoA-binding motif and is enzymatically inactive[21,22], and NatF is completely absent[13,17]. In humans, NatF is tethered to the Golgi membranes, where it specifically acetylates transmembrane proteins[17] and is thus likely to exhibit only a minor substrate overlap with NatC. Human Naa50 is catalytically active in its uncomplexed[62] and NatA-complexed form[21]. A comparison of the NatC substrate

pocket with those of human Naa50 and Naa60 reveals considerable differences in the shape, hydrophobicity, and electrostatic surface potential of the respective peptide-binding pockets (Supplementary Fig. 9). The NatC pocket, which is formed at the interface of subunits Naa30 and Naa35, is much deeper and more confined than the peptide-binding sites in NatE and NatF. While the NatC peptide-binding pocket is lined by several conserved hydrophobic residues, it still exhibits a moderate negative electrostatic surface potential as opposed to the positive charge in NatE. This may contribute to subtle differences in their substrate specificity. The unique substrate-binding pocket of NatC, including a substrate-binding interface in between the catalytic and auxiliary subunits, offers opportunities to design small molecules that specifically interfere with NatC function. Such compounds could be used to explore the cellular function of NatC in more detail. In light of the observation that NatC is upregulated in cancer cells[43], a therapeutic application of NatC inhibitors may also be envisioned.

Based on our mutagenesis data and previously proposed reaction schemes for GNATs[24,50,63] and NAT complexes, we present a refined catalytic mechanism for NatC (Fig. 4c). It was proposed for human Naa50 that a tyrosine and histidine cooperate in the deprotonation of the amino group via a coordinated water molecule[46], and a similar situation is found in Naa60 (refs. [47,49]). A glutamate was also suggested to function as the general base in the GNAT histone acetyltransferase Gcn5 from *S. cerevisiae*[64] and the *Salmonella typhimurium* GNAT RimI, which also employs a catalytic water[65]. An active site alignment of NAT orthologs reveals that the NatC catalytic subunit Naa30 has two potential general bases: Tyr80 and Glu118 (Fig. 4d, e). The proposed catalytic water is positioned 3.5 Å away from the peptide's amino group and is coordinated by the hydroxyl group of the Tyr80 side chain, the backbone carbonyl oxygen of Ile81 and the carboxyl group of the Glu118 side chain. E118A or E118Q substitutions resulted in a ~90% reduction, and Y80A or Y80F substitutions led to a ~50% reduction of the $k_{cat}$ compared to Naa30 WT (Table 2). This suggests that Glu118 serves as the key general base, while Tyr80 may support its function by coordinating the catalytic water. The deprotonated peptide amino group then performs a nucleophilic attack on the carbonyl carbon of the enzyme-bound acetyl-CoA, resulting in a transient zwitterionic tetrahedral intermediate. Simultaneously, the resultant negative charge on the carbonyl oxygen (oxyanion) may be stabilized by the backbone amide of Leu84. Tyr130 likely serves as a general acid that donates a proton to break the thioester bond of the tetrahedral intermediate. Y130A and Y130F mutants exhibited a strongly reduced $k_{cat}$, corresponding to ~1–2% of NatC WT turnover numbers, while showing no significant changes in $K_m$, emphasizing the crucial role of Tyr130 in catalysis. Tyr130 is universally conserved among NAT orthologs (Fig. 4d), and its role as a catalytic acid was initially proposed for *S. typhimurium* RimI[65]. Mutations in a corresponding tyrosine in NatB lead to similar reductions in $k_{cat}$ (ref. [50]).

After the nucleophilic attack, the nitrogen atom of the former peptide carries a positive charge and an excess hydrogen has to be removed in a second deprotonation step. Point mutations in the highly conserved Glu29 exhibited reductions of $k_{cat}$ to 4–6% of NatC WT turnover numbers, but the amino group of the substrate peptide and the carboxyl group of Glu29 are ~9 Å apart. Glu29 may play an indirect role in NatC catalysis, through direct contact with cognate NatC substrate peptides, resulting in a stabilization of active site loop α1–α2 and β6–β7. In NatA, mutations of the corresponding glutamate showed similarly strong reductions in $k_{cat}$ with negligible effects on $K_m$ (ref. [24]). However, a mutation of the corresponding glutamate in NatB increased its

enzymatic activity by ~60% (ref. [50]), pointing to alternative functions in different NATs.

A recent cryo-EM structure of the *S. cerevisiae* NatA/Naa50-ribosome complex revealed interactions with the ribosomal RNA near the peptide exit tunnel[66] (Supplementary Fig. 13a). To explore possible binding modes of NatC with the ribosome, the catalytic subunit of NatC was aligned with Naa10 in the NatA complex. In the resulting model, the auxiliary subunit Naa35 is in close contact with the surface of the ribosome. Moreover, the peptide-binding site is in close proximity to the ribosomal exit tunnel, and the Naa35 tip region contacts the ribosomal RNA, in agreement with its essential role in NatC–ribosome binding (Supplementary Fig. 13b, c). Mutations in the Naa35 tip region did not affect yeast growth on glycerol. However, the molecular basis for reduced growth of NatC deletion strains under non-fermentative conditions is not clear. It may be envisaged that the introduced mutations in the Naa35 tip only partially disturb the NatC–ribosome interaction in vivo. Alternatively, the NatC–ribosome interaction may not be required for the acetylation of all NatC substrates, including those implicated in yeast growth. Clearly, more work is required to fully understand the exact role of the co-translational activity of NatC in yeast and human.

Taken together, the structural and biochemical studies presented in this work provide insight into the unique architecture, substrate preference, catalysis, and ribosome interaction of NatC. They also show how sophisticated NAT machineries have divergently evolved to provide the cell with a broad repertoire of N-terminally acetylated proteins.

## Methods

**NatC expression and purification.** Genes encoding *S. cerevisiae* NatC subunits Naa30 (Uniprot ID: Q03503), Naa35 (Q02197), and Naa38 (P23059) were obtained from the Dharmacon yeast ORF collection and cloned into the pRSFDuet-1 (Novagen) expression vector (Supplementary Table 1). A NatC complex construct (designated NatC WT), expressing the truncated NatC subunits Naa30ΔC17 (residues 1–159), full-length Naa35 (residues 1–733), and Naa38ΔC11 (residues 1–77) was used for all kinetic and structural studies. The genes encoding Naa38ΔC11 and Naa30ΔC17 were cloned into the second multiple cloning site of the pRSFDuet-1 vector. An additional ribosome-binding site (AAGGAGATA-TACC) was added in front of the Naa30ΔC17 start codon. DNA encoding full-length Naa35, preceded by a human rhinovirus 3C (HRV 3C) cleavage site, was cloned in frame with the sequence for the His$_6$-tag in the first MCS. For NatC–ribosome co-sedimentation assays, an additional FLAG-tag (DYKDDDDK) was inserted in between the HRV 3C cleavage site and Naa35. NatC point mutant constructs were created by site-directed mutagenesis.

NatC expression vectors were transformed into *E. coli* BL21(DE3) cells, which were cultured in TB medium at 37 °C to an OD$_{600}$ of 0.6, followed by induction with 100 μM isopropyl β-D-1-thiogalactopyranoside and a temperature shift to 18 °C for overnight expression. Selenomethionine-substituted NatC complex was expressed in M9 minimal medium, supplemented with L-amino acids lysine, phenylalanine, threonine (100 mg l$^{-1}$), isoleucine, leucine, valine, and selenomethionine (50 mg l$^{-1}$)[67].

Cells were isolated by centrifugation and disrupted by a microfluidizer (Microfluidics) in lysis buffer (50 mM HEPES, pH 7.5, 150 mM NaCl, and 10 mM imidazole), supplemented with 1 μM DNAse (Roche), 1 mM MgCl$_2$, and 500 μM 4-(2-aminoethyl)-benzolsulfonylfluorid hydrochloride (BioChemica). Cleared lysates (~95,000 × *g*, 45 min, 4 °C) were incubated with 5 U mL$^{-1}$ Benzonase (Merck) for 30 min at 4 °C before application to a Ni$^{2+}$-sepharose column (GE healthcare). Following a wash step (20 mM HEPES, pH 7.5, 300 mM NaCl, and 25 mM imidazole), the NatC complex was eluted with 20 mM HEPES, pH 7.5, 150 mM NaCl, and 60 mM imidazole. Imidazole was removed with 50 kDa molecular weight cutoff concentrators (Amicon) and washing with size-exclusion buffer (20 mM HEPES, pH 7.5, and 150 mM NaCl). The solution was supplemented with 1:100 (w/w) GST-tagged HRV 3C protease, 5 U mL$^{-1}$ Benzonase and 1 mM MgCl$_2$, and incubated overnight at 4 °C. The mixture was loaded on a second Ni$^{2+}$ column and washed with size-exclusion buffer. The cleaved NatC complex was eluted with size-exclusion buffer containing 10 mM imidazole. Peak fractions were concentrated and loaded onto a Superdex200 column (GE) equilibrated with size-exclusion buffer. Fractions containing NatC were concentrated and flash-frozen in liquid nitrogen. A Naa38-deletion construct of NatC (ΔNaa38), FLAG-tagged NatC, and mutant NatC complexes were purified using the same protocol. The selenomethionine-substituted NatC complex was purified like the native NatC complex, except that 2.5 mM 2-mercaptoethanol was added to all buffers.

**Mass spectrometry.** Purified NatC complex was diluted in 0.1% trifluoroacetic acid (TFA) to a final concentration of 10 μM and mixed with an equal volume of saturated α-cyano-4-hydroxycinnamic acid solution in 50% acetonitrile/0.05% TFA. A total of 0.5 μL of the sample/matrix mixture was spotted on a MALDI target plate and measured with a mircoflex™ LRF MALDI-TOF mass spectrometer (Bruker) using linear, positive ion mode.

**NatC crystallization and structure determination.** NatC was diluted to 5–6 mg mL$^{-1}$ in size-exclusion buffer containing 2.5 mM 2-mercaptoethanol. The selenomethionine-substituted NatC complex was crystallized in its apo form. CoA was added at a 1:3 molar ratio to all native NatC proteins. In addition to CoA, the substrate peptides MFHLVGSRRR or MLRFVGSRRR were added to two further crystallization setups at a molar ratio of 1:5 or 1:3, respectively. Crystallization plates were set up as sitting drops in a 96-well format with a Crystal Gryphon dispensing robot (Art Robbins Instruments) by mixing 0.2 μL protein solution with 0.2 μL reservoir solution above an 80-μL reservoir. Reservoir solution contained 14.5–16.5% PEG 4000, 150 mM ammonium iodide, and 100 mM sodium citrate, pH 6.1–6.3. Diffraction quality crystals appeared after 12–48 h and required another 5 days to reach maximum dimensions. Crystals were soaked for ~10 s in well solution containing 20% (v/v) ethylene glycol and quickly frozen in liquid nitrogen. All datasets were collected at beamline BL14.1 at BESSY II in Berlin at a temperature of 100 K.

The dataset of the selenomethionine-substituted NatC complex (NatC, apo) was taken from a single crystal at a wavelength of 0.9797 Å (3600 images, 0.1° per frame) and processed to 2.40-Å resolution using the XDS suite[68]. Experimental phasing was achieved by SAD phasing with SHELXC/D/E[69], using HKL2Map[70] from the CCP4 interface[71]. An automated model building and refinement was performed with Buccaneer[72], and the model was completed manually in Coot[73] and refined in Phenix[74]. Translation–libration–screw-rotation (TLS) refinement employed two TLS groups for subunit Naa30ΔC17, two groups for Naa38ΔC11 and five groups for subunit Naa35.

Datasets of native, ligand-bound NatC complexes were taken from a single crystal each, at a wavelength of 0.9184 Å and consisted of 1100 images (NatC•CoA•MFHLV) or 1800 images (NatC•CoA and NatC•CoA•MLRFV) with an oscillation of 0.1° per frame. All native datasets were processed with XDS to 2.45-Å (NatC•CoA), 2.99-Å (NatC•CoA•MFHLV), or 2.75-Å (NatC•CoA•MFHLV) resolution, respectively. Datasets were phased by molecular replacement with Phaser[75] using the refined structure of the SeMet-labeled NatC complex, followed by several rounds of manual model building (Coot) and refinement (Phenix). All ligand occupancies were refined. TLS was used in later stages of the refinement using the same TLS groups as for the selenomethionine dataset plus one additional group for CoA and another for the peptide ligand. Simulated annealing OMIT maps for CoA, substrate peptides and a water molecule in the active site were generated with phenix.maps after performing a simulated annealing refinement run in Phenix. Figures were prepared with the PyMOL Molecular Graphics System, Version 2.3.1 (Schrödinger, LLC). Secondary structure assignment for all NatC structures was done manually. NatC dimension were calculated with the PyMOL script "Draw Protein Dimensions" (https://pymolwiki.org/index.php/Draw_Protein_Dimensions). Interfaces and buried surface areas between NatC subunits were calculated using the PDBe PISA webserver[76]. NatC subunit structures were compared against the PDB database, using the DALI webserver[77]. Structural superpositions and RMSD calculations were performed with PyMOL. The electrostatic surface potential and surface hydrophobicity were calculated with the PyMOL plugins APBS[78] or VASCo[79]. Multiple sequence alignments for Naa30, Naa35, and Naa38 were generated with Geneious (https://www.geneious.com/) and visualized using ESPript3.0 (http://espript.ibcp.fr/). Sequence logos of the catalytic subunits of five different NAT paralogs (Naa10, Naa20, Naa30, Naa50, and Naa60) were generated from individual multiple sequence alignments using the corresponding NAT orthologs from 18 different species: *S. cerevisiae, Kluyveromyces lactis, Candida albicans, Fusarium graminearum, Physcomitrelle patens, Oryza sativa, Zea mays, A. thaliana, Populus trichocarpa, Ricinius communis, Drosophila melanogaster, Aedes aegypti, Tribolium castaneum, Strongylocentrotus purpuratus, Danio rerio, Xenopus tropicalis, Homo sapiens,* and *Mus musculus.*

**Acetyltransferase assays.** The acetyltransferase activity was determined using the Ellman method, adapted from Thompson, et al.[80]. Reactions were carried out at 25 °C in acetylation buffer (50 mM HEPES, pH 7.5, 150 mM NaCl, 0.2 mM EDTA, and 0.003% (v/v) Tween-20). Tween-20 was added to the buffer to reduce protein surface adsorption. Catalytic parameters for NatC WT and mutants were determined using the yArl3 peptide (MFHLVGSRRR), containing the five N-terminal residues of the *S. cerevisiae* ADP-ribosylation factor-like protein 3 (Uniprot ID Q02804), followed by a GS linker and a triple arginine to facilitate peptide solubility. Reactions were carried out as duplicates at six different peptide concentrations and an acetyl-CoA concentration of 500 μM ($\geq 10 \times K_m$). Individual reactions (110 μL total volume) were performed with final NatC concentrations between 50–2500 nM for 150 s for NatC WT and up to 100 min for catalytically impaired NatC mutant complexes. At regular time intervals, 20-μL aliquots were taken from the reaction and quenched in 40 μL quenching buffer (3.2 M guanidinium-HCl, 5 mM EDTA, 100 mM sodium phosphate, pH 6.8), containing freshly added 5,5-dithiobis(2-nitrobenzoic acid) (DTNB) at a final concentration of 0.5 mM. A total

of 50 μL of the quenched reactions were transferred into 384-well plates and absorbances were measured at 412 nm with a M1000 Pro Microplate reader (Tecan). TNB$^{2-}$ anion product concentration, was determined using the Beer–Lambert law ($A = \varepsilon \times c \times l$), assuming $\varepsilon = 14{,}150\,M^{-1}\,cm^{-1}$ (ref. [81]). Reaction background absorbances (containing quenched enzyme) were determined for each reaction and subtracted from the absorbances of the individual reactions. Plate background absorbances were subtracted from all reactions to account for plate-specific imperfections. Turnover of limiting substrate did not exceed 20%. Initial velocities were fitted by nonlinear least fit squares to the Michaelis–Menten equation ($v_0 = v_{max}[S]/(K_m + [S])$) using GraphPad Prism version 6.01 to determine $k_{cat}$ and $K_m$ parameters. Each complete acetylation assay was performed at least in triplicate, and the average $k_{cat}$ and $K_m$ and SD were calculated with Microsoft Excel 365. The $K_m$ of acetyl-CoA for NatC WT was determined with a fixed Arl3 peptide concentration of 1340 μM and 30–500 μM acetyl-CoA. A NatA substrate peptide (SASEAGSRRR) containing the N-terminal five residues (after removal of the initiator methionine) of the *S. cerevisiae* ThrRS (Uniprot ID: P04801); and a NatB substrate peptide (MDSEVGSRRR) from *S. cerevisiae* actin (Uniprot ID: P60010) showed no activity, even after incubating for 2 h and using a large excess (2 μM) of NatC WT. The Gag peptide (MLRFVGSRRR), containing the N-terminal five residues of the major capsid protein (Gag) of the *S. cerevisiae* virus L-A (Uniprot ID: P32503) showed a higher specificity constant, and thus the acetyltransferase assay was carried out: final NatC concentrations: 10–80 nM; total reaction times: as short as 60 s; peptide concentrations: 30–500 μM; total reaction volume: 226 μL, with larger aliquots of 72 μL taken. Aliquots were quenched in 36 μL of 2× quenching buffer (6.4 M guanidinium-HCl, 10 mM EDTA, 200 mM sodium phosphate, pH 6.8). Two times 50 μL of each quenched reaction were transferred into separate microplate wells to obtain duplicate absorbance readings, which were averaged. Further peptides, exhibiting a $K_m > 50$ μM, were measured as described for the yArl3 peptide. Peptides exhibiting a $K_m < 50$ μM were measured as described for the Gag peptide.

All peptides were synthesized by ProteoGenix (Schiltigheim, France) with a purity of ≥95% and with TFA exchanged for HCl. Peptides were solubilized in acetylation buffer and peptide concentrations were determined photometrically from TNB$^{2-}$ product concentration, formed in an end point acetylation reaction. For each peptide stock, five different dilutions were incubated together with an excess of CoA and 2 μM NatC WT in acetylation buffer. Reaction aliquots were quenched after 1.5 and 2 h with a twofold excess of quenching buffer, containing 0.5 mM DTNB. Measurements taken after 2 h did not show a significant increase of absorbance compared to the 1.5 h time point and both readings were within 5% of one another.

**Purification of yeast 80 S ribosomes.** Yeast 80 S ribosomes were purified according to a protocol adapted from Magin, et al.[57]. *S. cerevisiae* strain YA2488 was plated out on YPD-agar plates (1% yeast extract, 2% peptone, and 2% glucose), and used to inoculate a 20 mL preculture of YPD, which was incubated at 30 °C for 7 h. A volume of 8 L of YPD was inoculated to an OD$_{600}$ of 0.001 and grown at 30 °C to an OD$_{600}$ of 2 (stationary phase). Cells were pelleted at 8000 × *g*, 5 min, 4 °C (Beckman Rotor JLA9.1000), resuspended in YP medium (1% yeast extract, 2% peptone, and no glucose), and incubated for 10 min, 250 r.p.m., 30 °C, to ensure that all ribosomes would be in the "apo" form, without nascent chain or tRNA. The following cell lysis steps were all carried out at 4 °C: cells were pelleted by centrifugation at 8000 × *g*, 5 min (Beckman Rotor JLA9.1000) and resuspended in buffer A (30 mM HEPES, pH 7.5, 50 mM KCl, 10 mM MgCl$_2$, 8.5% sorbitol, 2 mM DTT, and 0.5 mM EDTA). Cells were centrifuged again at 8000 × *g*, 5 min, the pellet was weighed, and resuspended in lysis buffer (buffer A + one cOmplete mini EDTA-free protease inhibitor tablet (Roche 11836170001), 1 unit μL$^{-1}$ RNAsin (N261B, Promega), and 800 μg mL$^{-1}$ heparin) to a final concentration of 200% (w/v). Glass beads (G8772 sigma) were added to the resuspension and lysis was performed by vortexing for 30 s, followed by 30 s on ice, repeated four times. To obtain the yeast lysates, the resuspension was centrifuged at 9000 × *g*, 10 min (rotor Beckman TA-10.250), and the supernatant was collected. The absorbances at 260 and 280 nm were measured. To isolate the 80 S monosomal ribosome fraction, yeast lysate ($A_{260} = 200$) was underlaid with 1 mL 30% sucrose in 80 S buffer (20 mM HEPES, pH 7.6, 100 mM potassium acetate, 5 mM MgCl$_2$, and 2 mM DTT) and centrifuged 70,000 × *g*, 18 h (Beckman ultracentrifuge, rotor mla80). The pellet was collected, and added at a concentration of $A_{260} = 100$, to the top of a 15–30% sucrose gradient, which was prepared by underlying 15% sucrose in 80 S buffer with 30% sucrose in 80 S buffer, and mixing (Gradient Master, BIOCOMP, short settings). The gradients were then spun at 92,703 × *g*, 6 h, using a SW32 Ti Beckman swing-out rotor, and then loaded to a polysome collector (LKB). Fractions containing 80 S monosomes were pooled and centrifuged at 127,959 × *g*, 14 h (Beckman MLA-80 rotor). The pellet containing the 80 S yeast ribosomes were resuspended in 20 mM HEPES pH 7.6, 50 mM potassium acetate, 5 mM MgCl$_2$, and 2 mM DTT to give a final concentration of 2 μM.

**NatC–ribosome sedimentation assay.** The sedimentation assay was adapted from the protocol described by Magin et. al.[57]. NatC and ribosomes were combined to final concentrations of 1 μM and 0.8 μM, respectively in 80 S buffer (20 mM HEPES, pH 7.5, 50 mM KOAc, 5 mM Mg(OAc)$_2$, 2 mM DTT, and 0.003% (v/v) Tween-20). Tween-20 was added to reduce unspecific binding of NatC to plastic surfaces. A volume of 35 μL of the NatC/ribosome solution was underlaid with

80 μL of 80 S buffer supplemented with 30% (w/v) sucrose. Samples were centrifuged for 70 min at 352,769 × *g* in a precooled (4 °C), TLA-100 rotor (Beckman). The supernatant (i.e., 115 μL) was carefully removed and mixed with 30 μL of 5× SDS sample buffer (250 mM TRIS-HCl, pH 6.8, 10% (w/v) SDS, 30% glycerol, 10% (v/v) 2-mercaptoethanol, and 0.5% (w/v) bromophenol blue). The pellet was resuspended in 145 μL resuspension buffer (35 μL 80 S buffer, 80 μL 80 S buffer with 30% sucrose, and 30 μL 5× SDS sample buffer), and run on an SDS–PAGE gel to analyze the amount of NatC and ribosome in the supernatant and pellet fractions by western blot, using mouse α-FLAG M2 (Sigma-Aldrich, F3165, 1:10,000 dilution) and α-uL30 (α-RPL7, Abcam, ab72550, 1:3,000) as primary antibodies, and horseradish peroxidase-coupled goat α-mouse IgG (Dianova, 115-035-003, 1:10,000) and α-goat IgG (Dianova, 111-035-003, 1:10,000) as secondary antibodies. Chemoluminescence intensities of protein bands were analyzed with ImageJ ImageJ 1.52a. Sedimentation assays were carried out in triplicate for NatC WT and each mutant NatC construct.

**Yeast genetics.** *S. cerevisiae* WT (BY4741) and *Naa35*-deletion (Y00294) strains were obtained from Euroscarf. The *NAA35* gene (YEL053C), including 460 bp upstream and 494 bps downstream of the *NAA35* ORF, was amplified by PCR from *S. cerevisiae* genomic DNA (strain BY4741), and cloned into the BamHI cloning site of the pRS416 yeast centromere vector. Plasmids encoding Naa35-mutants Tip1 (K500A, K501A, K503A, and K504A) and Tip2 (K511, R515, and R519) were created using overlapping fragments, containing the desired mutations. Additional 27 bases (ATGGATTATAAAGATGATGATGATAAA) encoding for an additional FLAG-tag were inserted in front of the start codon of *NAA35*-WT and mutant genes for immunodetection of the *NAA35* gene products. For vector transformations, *S. cerevisiae* WT (BY4741) and *NAA35*-deletion (Y00294) strains were grown overnight in 5 mL YPD (10 g L$^{-1}$ yeast extract, 20 g L$^{-1}$ peptone, and 2% (w/v) glucose) medium at 30 °C. Cells were sedimented at 3000 × *g*, washed with 1 mL of LiOAc mix (100 mM lithium acetate, 50 mM EDTA, 100 mM Tris-HCl, pH 7.6), and resuspended in 100 μL LiOAc mix. To this, 10 μL of freshly boiled herring sperm DNA (Sigma), 1 μg of plasmid DNA, and 700 μL of PEG mix (40% (w/v) PEG 4000, 100 mM lithium acetate, 50 mM EDTA, 100 mM Tris-HCl, pH 7.6) was added and the suspension was incubated for 30 min at 30 °C, followed by addition of 48 μL DMSO and 15 min incubation at 42 °C. After addition of 3 mL YPD, cells were grown for 1 h at 30 °C and plated on SD-ura (6.7 g L$^{-1}$ yeast nitrogen base, 1.92 g L$^{-1}$ SD medium supplement without uracil) agar plates containing 2% (w/v) glucose, which were grown for 2–3 days at 30 °C. FLAG-tagged Naa35 could not be detected by western blot, likely due to low Naa35 expression levels from the endogenous promotor, as previously suggested[58].

**Yeast dilution spot assays.** Single colonies of transformed *S. cerevisiae* WT and *NAA35*-deletion strains were used for overnight cultivation at 30 °C in SD-ura medium, containing 2% (w/v) glucose. After 16 h, cells were sedimented (5 min, 3000 × *g*) and resuspended in SD-ura medium, containing 3% (v/v) glycerol to an OD$_{600}$ of 1. Cells were grown at 30 °C for 3 h to deplete remaining glucose. For dilution spot assays, yeast cells were initially diluted to an OD$_{600}$ of 1. Subsequent tenfold dilutions were made and a 48-pin multi-blot replicator was used to spot cells onto SD-ura agar plates containing 2% (w/v) glucose or 3% (v/v) glycerol, respectively. Plates were incubated at 37 °C for 3–5 days.

**Statistics and reproducibility.** Statistical parameters including the definitions and exact value of *n*, deviations, *p* values, and the types of the statistical tests are reported in the figures and corresponding figure legends or hereafter. The SDS–PAGE analysis in Fig. 1d was repeated twice with identical results. The underlying NatC complex purifications were performed three times for the NatC complex containing full-length Naa35 (residues 1–733) and twice for the NatC complex containing Naa35ΔN17 (residues 18–733). Replicate purifications showed identical results. The NatC complex containing Naa35ΔN44 (residues 44–733) was purified once. Statistical analyses were carried out using GraphPad Prism version 6.01.

**Reporting summary.** Further information on research design is available in the Nature Research Reporting Summary linked to this article.

## Data availability

The atomic coordinates of NatC have been deposited in the Protein Data Bank with accession numbers 6YGA (NatC, apo), 6YGB (NatC•CoA), 6YGC (NatC•CoA•MFHLV), and 6YGD (NatC•CoA•MLRFV). The authors declare that the main data supporting the findings of this study are available within the article and its Supplementary Information files. All other supporting data, plasmids, and yeast strains developed in this study are available from the corresponding author upon reasonable request. Source data are provided with this paper.

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

## Acknowledgements

This project was supported by grants from the Deutsche Forschungsgemeinschaft (SFB958/A12 to O.D. and C.M.T.S.). We thank the entire BESSY team for generous support during data collection at beamline MX14.1. We also thank Florian Wollweber and Martin van der Laan (Center for Molecular Signaling, Saarland University) for providing a yeast expression vector and useful advice on yeast genetics. We furthermore thank Christian Lips, Ernst Jarosch, and Thomas Sommer (Max-Delbrück-Center for Molecular Medicine) for their advice and practical help on the yeast spot dilution assay. We thank Stephen F. Marino and Udo Heinemann for critical reading of the manuscript.

## Author contributions

S.G. designed the constructs, determined the structures, developed and performed all biochemical assays, and performed yeast experiments, if not otherwise noted. L.V.M.H. contributed to protein purifications and activity measurements. T.B.-B. assisted in structure solution and the development of the reaction scheme. C.C.M.L. assisted with the ribosome co-sedimentation assays. C.M.T.S. and O.D. supervised research. S.G. and O.D. wrote the manuscript and all authors edited the manuscript.

## Funding

## Competing interests

The authors declare no competing interests.
