## [Peer Review File · Nature Communications]

REVIEWER COMMENTS

Reviewer #1 (Remarks to the Author):

The manuscript "Divergent architecture of the heterotrimeric NatC complex explains N-terminal acetylation of cognate substrates" by Stephan Grunwald et al. presents the crystal structure and biochemical mechanism of this eukaryotic NAT complex for the first time.

The study represents a conceptual advance in our understanding of how a major eukaryotic protein modification is carried out by N-terminal acetyltransferase enzymes. The structural determination of the NatC complex has remained unsolved, and this study points to several unique features of this enzyme (as compared to the other NATs) that allow us to understand its substrate specificity at the atomic level.

The experiments are technically sound and support the conclusions presented.

The manuscript is well written and properly contextualizes the findings with current knowledge in the field.

Suggestions for revisions:

1. Lines 36-37: '.which act co-translationally on nascent polypeptide chains, as they emerge from the ribosomal exit tunnel [15-17].'

-> References 15 and 16 (Polevoda et al., 2008; 2009) are suitable, but I would replace Ref. 17 (Hole et al., 2011) with Gautschi et al., Mol Cell Biol, 2003, PMID:14517307. Then the yeast NATs are properly covered for this statement. For human NATs a separate series of papers would have to be cited.

2. Lines 46-47: 'NatB acetylates N-termini starting with methionine, followed by a residue with an acidic or amide side chain [27, 28].'

-> References used are suitable, but I would additionally add Polevoda et al., EMBO J, 1999 (ref. [29]) to this statement.

3. Lines 49-51: 'The substrate specificity profiles of NatE and NatF partially overlap with that of NatC, and together they Nt-acetylate ~28% of the human proteome [3, 9, 22, 36].'

-> This is correct and references are suitable, but to avoid confusion for the reader, please clarify that it is the in vitro specificity that overlaps (and when assessed ectopically in yeast). Most likely these three NATs have mostly non-overlapping substrates in vivo due to their differential profile regarding functional complexes and subcellular localization.

(also expand a bit on this in the Discussion regarding the in vivo overlap in substrate Nt-acetylation (or lack thereof) in yeast and humans: yeast: only Naa30 and Naa50 present and yeast Naa50 might not be catalytically active thus perhaps leaving all substrates of this type to Naa30; humans: NAA60 targeting transmembrane proteins from its Golgi-membrane position, NAA50 catalytically active, but associated with NatA components, while NAA30 is in the NatC complex)

4. Lines 62-63: 'The wide range of biological processes that contain targets for NatC-mediated Nt-acetylation, make it essential for proper cell growth and development, across all biological kingdoms.'

-> Please rephrase. Does NatC really exist in all biological kingdoms?

5. Lines 113-115: 'Naa30 is most similar to the archaeal NAT ortholog Ard1 and the Naa10 subunit of the Schizosaccharomyces pombe NatA complex, respectively (Extended Data Fig. 3b).'

-> Please insert appropriate references to these structures.

6. Lines 201-203: 'However, such a substrate is unlikely to exist in vivo, as aminopeptidases usually cleave the amino-terminal methionine when it precedes a small amino acid [52).'

-> This is correct unless there are inhibitory residues close to the N-terminus (inhibitory for MetAP action). For instance, the presence of Pro at the third position (Met-Xxx-Pro) will potentially inhibit MetAP activity (Xiao et al., Biochemistry, 2010, PMID:20521764).

Such N-termini are also acetylated in cells, for instance several Met-Ala-Pro are found to be acetylated by NatF (Aksnes et al., Cell Rep, 2005, PMID:25732826).

7.

- Lines 190-191: 'In both structures, the N-terminal methionine is sandwiched between Leu27 and Tyr145 of Naa30...'

- Lines 313-314: 'In both cases, Met1 is sandwiched in between Tyr145 and Leu27.'

-> Met1 from Gag peptide is stacked between Leu27 and Tyr145, but for Arl3 peptide Leu27 side chain is flipped according to the figures 3c, 3d and 4a. This is not clear from the text and should be clarified, especially as the authors also state that this is an important residue for catalysis.

8.

- Lines 242-243: 'At peptide position 3, amino acids with a small or elongated side chain are favored, whereas a bulky amino acid appears to reduce binding'

- Lines 321-322: 'However, a histidine, or, presumably, other bulky amino acids at position 3, hinder catalytic activity.'

-> The lowest Km reported in this article is for a peptide that has an Arginine at this position. Arginine is certainly bulkier than histidine, so another way to phrase this should be found.

9. Table 2, Figure 4b.

-> Specificity constant (kcat/Km) should not be used to compare different enzymes (or variants of the same enzyme), see e.g. PMID: 17433847. Separate plots presenting kcat and Km would also be more informative.

Reviewer #2 (Remarks to the Author):

NatC is a key cellular enzyme, which acetylates the N-terminus of cytosolic proteins starting with a Methionine followed by a hydrophobic or amphipathic amino acid, with amino acid in positions 3 and 4 contributing to the selectivity of NatC.

This is a very interesting paper considering the fundamental role of N-terminal protein acetylation by NatC in many aspects of cells and diseases, and the lack of information on its mechanism and structure.

The paper by Grunwald et al., reports several crystal structures of the Saccharomyces cerevisiae NatC complex in the apo for CoA-bound form and two different N-terminal peptide-substrates

(MFHLV and MLRFV). In the various solved structures, NAA30 adopts the typical GNAT fold, while the small auxiliary subunit NAA38 adopts a Sm fold but without the Lsm-IRG motif that mediates the association with small nuclear RNAs.

Even more interestingly, the authors reveal that NAA35 displays a never seen fold, acting as a central assembly hub for NAA30 and NAA35.

Strikingly, comparison of NatC with NatA and NatB complexes shows remarkable differences. Based on ligand-induced structural conformational changes of NAA30 and activity characterization, the authors propose a model of catalytic mechanism for the NatC complex.

Finally, the authors identify an electropositive region of NAA35 necessary for the association of NatC with ribosome and propose a NatC-ribosome interaction model.

This work could significantly contribute to the knowledge of Nat enzymes and allow a significant step forward in the field of N-terminal acetylation process, providing mechanistic insights on one of the major Nats.

However, I think that the paper would be strengthened by addressing several issues described below.

Major points

1/ One of my major concerns is related to the NatC-substrate selectivity characterization.

Although, the authors have tested several peptides variants, several specific substitutions are missing to fully understand the NatC selectivity. For instance, studies on the effects on the catalytic parameters of a non-favored residue in the position 2, such as Asp/Glu or Arg, in a yAr13 (F2D/E/R) or in peptide variants of the protein Gag (L2D/E/R), are missed. This information would certainly clarify the role of residues at position 3,4 and 5 in the selectivity of NatC. Likewise, no unfavorable substitution at position 4 or 5 has been studied and discussed by the authors, while these specific positions seem to be involved in the formation of the main hydrophobic pocket, important for selecting hydrophobic residues at position 2 in NatC.CoA.MFHLV or NatC.CoA.MLRFV complexes. Likewise, no substitution as F4G with an unfavorable residue at position 4 of the Gag peptide was attempted to confirm the importance of the extended pocket in NatC, as stated by the authors.

I suggest that the authors complete their selective characterization, by monitoring the catalytic activity of NatC with additional peptides variants according to what is described above.

2/ In lines 230-240, after comparison with Naa10 of the NatA complexes and the hypothesis that the binding of the substrate has no impact on the catalysis of Naa10, the authors suggest that the NatC.CoA.MLRFV (Gag) structure represents an efficient catalytic conformation of the NatC.. Does this mean that NatC.CoA.MFHLV (yAr13) does not represent a catalytic efficient conformation of NatC, which looks more like an apo conformation?

The authors analyzed the different amino acid substitutions using the structure of NatC.CoA.MFHLV as reference. Instead, I would suggest analyzing these substitutions using the NatC.CoA.MLRFV.

3/ In lines 241-243, the authors claimed that "Based on the presented data, NatC seems to prefer methionine-starting substrate peptides with hydrophobic amino acids at position 2 and 4. At peptide position 3, amino acids with a small or elongated side chain are favored, whereas a bulky amino acid appears to reduce binding". How authors can conclude based on their data that hydrophobic amino acids at position 2 and 4 are favored when no hydrophilic substitution was tried at these positions?

In light of the above, the conclusion drawn on the impact of position 3 "At peptide position 3, amino acids with a small or elongated side chain are favored, whereas a bulky amino acid appears to reduce binding" makes no sense to me. The authors should improve this part with further experiments as proposed in point 1 and re-write the text accordingly.

4/ The authors, using co-sedimentation assay, identified EPR2 Tip/Tip2 as one of the major elements involved in NatC-ribosome bond. Since the presence of NAA35 is necessary to restore NAA30 activity in vivo, I suggest the authors to complement the yeast KO NAA35 mutant with the NAA35-EPR2 variant used in the sedimentation assay. This will provide information if EPR2 alone is

sufficient to ensure the NatC-ribosome binding.

5/ Considering the proposed mechanism for NatC, the authors proposed that Glu29 might be in charge of the second phase of deprotonation p . However, none of their structural data is in favor of this suggestion. Indeed, the authors recognized that the substrate peptide amino group and the carboxyl group of Glu29 are 9 Å away in NatC•CoA•MLRFV and therefore suggested that Glu29, being part of the flexible $\alpha 1$ - $\alpha 2$ loop, may relocate to the active site. However the authors previously suggested in the manuscript that NatC•CoA•MLRFV was in the efficient catalytic conformation. This is not in accordance with a possible movement of the flexible $\alpha 1$ - $\alpha 2$ loop. Furthermore, Glu29 is not conserved in different Nats (Fig.4d), in contrast to the other key residue Y130 (whose key function was first identified by Vetting 2008 in RimI). In their structures Fig3, Glu29 interacts either with the substrate peptide of residue 3 or with the backbone of residue 4. Since Glu29 could be the involved catalytic base and/or involved in binding the substrate and is far away from its proposed function, it would be better to remove the role of Glu29 in Fig. 4c.

Minor points

1/ NAA50 is not active in yeast and recently (Knorr NSMB 2019, Deng 2019, 2020) it was suggested to act as an auxiliary subunit (at least in yeast). Please, clarify this point in the introduction (lines 41-42).

2/ Taking into account the recent Arnesen's review, NatC/E/F in human is estimated around 14.4% and 9.6% in yeast. Please, adjust the value (28%) reported in line 50.

3/ The authors clearly show remarkable differences in the tertiary and quaternary structures of NatC in comparison to those of NatA and NatB. In this context, the authors claimed that $\beta 6\beta 7$ element of the catalytic subunit NAA30, which builds the C-terminal half of the active site of NatC, is not motionless as in NatA or NatB complexes. Indeed, this element is constrained by NAA35 subunit in NatC. Similarly, the remaining N-terminal half of the active site, made of $\alpha 1\alpha 2$ loop, is stabilized differently in NatC than in the NatA or NatB complexes. However, the authors conclude this comparison with the following sentence (line 163-165) "Thus, while the overall architecture of the auxiliary subunits is different in the three NAT complexes, similar mechanistic principles apply for the stabilization of the catalytic center". This last sentence of the paragraph should be changed because it is not in agreement with the whole paragraph and Figure 2. Alternatively, the authors should better explain what are the essential stabilizing elements, similar to those of NatA and NatB, found in NatC catalytic center which allow to suggest a similar mechanistic principle.

4/ At line 200-2003, the authors wrote "This demonstrates that a methyl side chain at position 2 is, in principle, sufficient to mediate binding. However, such a substrate is unlikely to exist in vivo, as aminopeptidases usually cleave the amino-terminal methionine when it precedes a small amino acid [52]"

I suggest that this part should be re-written taken into account the fact that the specific MAP (Proline at position 3) is a non-optimal MetAP substrate and very often Methionine is not cleaved.

5/ Fig. 3b is very difficult to understand. The top view they choose makes it difficult to apprehend the differences between both substrates conformation within the active site and particularly at position 3, 4, 5, highlighted as important for substrate recognition and for which the omit map is less clear. I suggest the authors provide a different view for Fig3b (perhaps side view) and to divide the figure in two parts, one centered on the substrate and the other on the CoA. In addition, a superposition between the two substrates would be useful to understand their binding differences in the active site and values in Table2. As well a superposition between the two CoA in a different view will help to understand the different conformation observed in the structure of both peptide substrates as observed in Fig. 3d.

6/ The section on how NatC interacts with ribosome and the logic used is extremely dangerous and misleading (lines 264-276). The quaternary arrangement of NatC and NatA/E is completely different and currently no information is available on a possible exclusive interaction of these complexes on the ribosomal surface. Therefore, I do not understand the logic to locate NAA30 in an equivalent position of NAA10. It is not surprising to see that the peptide-binding site of NAA30

is therefore found near the ribosomal exit tunnel when NAA30 is aligned on NAA10. Without further results, these models make no sense and this paragraph should be removed from the result section. The authors may discuss this part in the discussion section, but soften their conclusions by stating that additional structural and functional studies are needed to confirm them. 7/The authors identified Glu118 as the catalytic base with support of Tyr80 but do not discuss Glu29, the replacement of which with Ala or Gln also inhibits NatC activity. Moreover, the equivalent Glu29 residue in NAA10 was previously identified as the catalytic base of NatA. This should be discussed.

We would like to thank both referees for their positive and very constructive feedback, which led to significant improvements of our manuscript. Based on their suggestions, we performed several new experiments and modified the manuscript at several positions, as indicated in the detailed point-by-point response and marked in red in the revised version of the manuscript. In line with the new format guidelines of Nature Communications, the following changes were introduced to figures and supplementary figures:

Fig. 1d shows the various NatC variants on a single SDS-gel.

Fig. 3a includes kinetic results from further peptides. The y-axis is now labelled 'Specificity constant'. For better comparison with previous literature, it is now shown as $\times 10^{-3} \text{ s}^{-1} \mu\text{M}^{-1}$.

Fig. 3b shows an overlay of the two peptides in the peptide binding pocket.

The electron density maps of old Fig. 3a have been moved to Supplementary Fig. 8a.

Fig. 4b now shows the k_{cat} instead of the $k_{\text{cat}}/K_{\text{m}}$.

Fig. 4b and 5c: The individual data points are now included in the bar graphs.

Fig. 4c: Glu29 was removed.

Fig. 5d contains new yeast spot dilution assays.

Supplementary Fig. 1 contains the sequence alignment of Naa30 and Naa38 orthologues.

Supplementary Fig. 2 contains the Naa35 sequence alignment.

Supplementary Fig. 3 describes the purification and mass spec analysis of the NatC complex.

Supplementary Fig. 6 contains the Michaelis-Menten kinetics of NatC for various peptides.

Supplementary Fig. 7 contains the Michaelis-Menten kinetics of various NatC mutants.

Supplementary Fig. 8 assembles various electron density maps.

Supplementary Fig. 9 contains an additional NatC surface representation with the γArl3 peptide.

Supplementary Figure 10c was minorly modified to better illustrate peptide-induced rearrangements.

Supplementary Figure 11 contains the replicates of the ribosome co-sedimentation assay.

Supplementary Figure 12 contains the replicates of the yeast growth assay.

Supplementary Figure 13 has been simplified.

Table 2 has been expanded to include kinetic data from new peptides. The sample number n was removed and instead, individual graphs are shown in the corresponding Supplementary Figures 6 and 7.

REVIEWER COMMENTS

Reviewer #1 (Remarks to the Author):

The manuscript "Divergent architecture of the heterotrimeric NatC complex explains N-terminal acetylation of cognate substrates" by Stephan Grunwald et al. presents the crystal structure and biochemical mechanism of this eukaryotic NAT complex for the first time.

The study represent a conceptual advance in our understanding of how a major eukaryotic protein modification is carried out by N-terminal acetyltransferase enzymes. The structural determination of the NatC complex has remained unsolved, and this study points to several unique features of this enzyme (as compared to the other NATs) that allow us to understand its substrate specificity at the atomic level. The experiments are technically sound and support the conclusions presented. The manuscript is well written and properly contextualizes the findings with current knowledge in the field.

Thank you very much!

Suggestions for revisions:

1. Lines 36-37: ‘..which act co-translationally on nascent polypeptide chains, as they emerge from the ribosomal exit tunnel [15-17].’

-> References 15 and 16 (Polevoda et al., 2008; 2009) are suitable, but I would replace Ref. 17 (Hole et al., 2011) with Gautschi et al., Mol Cell Biol, 2003, PMID:14517307. Then the yeast NATs are properly covered for this statement. For human NATs a separate series of papers would have to be cited.

As suggested, we replaced Hole et al. with Gautschi et al. (lines 36).

2. Lines 46-47: ‘NatB acetylates N-termini starting with methionine, followed by a residue with an acidic or amide side chain [27, 28].’

-> References used are suitable, but I would additionally add Polevoda et al., EMBO J, 1999 (ref. [29]) to this statement.

The recommended reference was added (line 44).

3. Lines 49-51: ‘The substrate specificity profiles of NatE and NatF partially overlap with that of NatC, and together they Nt-acetylate ~28% of the human proteome [3, 9, 22, 36].’

-> This is correct and references are suitable, but to avoid confusion for the reader, please clarify that it is the in vitro specificity that overlaps (and when assessed ectopically in yeast). Most likely these three NATs have mostly non-overlapping substrates in vivo due to their differential profile regarding functional complexes and subcellular localization.

We added a corresponding statement to the introduction (line 47-48).

Also expand a bit on this in the Discussion regarding the in vivo overlap in substrate Nt-acetylation (or lack thereof) in yeast and humans: yeast: only Naa30 and Naa50 present and yeast Naa50 might not be catalytically active thus perhaps leaving all substrates of this type to Naa30; humans: NAA60 targeting transmembrane proteins from its Golgi-membrane position, NAA50 catalytically active, but associated with NatA components, while NAA30 is in the NatC complex.

We added the corresponding section to the discussion (lines 321-326).

4. Lines 62-63: ‘The wide range of biological processes that contain targets for NatC-mediated Nt-acetylation, make it essential for proper cell growth and development, across all biological kingdoms.’

-> Please rephrase. Does NatC really exist in all biological kingdoms?

Indeed, NatC is indeed only present in eukaryotes (line 61).

5. Lines 113-115: ‘Naa30 is most similar to the archaeal NAT ortholog Ard1 and the Naa10 subunit of the Schizosaccharomyces pombe NatA complex, respectively (Extended Data Fig. 3b).’

-> Please insert appropriate references to these structures.

We added references (Liszczak, Glen et al., 2013) and (Liszczak, Goldberg et al., 2013) (line 112).

6. Lines 201-203: ‘However, such a substrate is unlikely to exist in vivo, as aminopeptidases usually cleave the amino-terminal methionine when it precedes a small amino acid [52].’

-> This is correct unless there are inhibitory residues close to the N-terminus (inhibitory for MetAP action). For instance, the presence of Pro at the third position (Met-Xxx-Pro) will potentially inhibit MetAP activity (Xiao et al., Biochemistry, 2010, PMID:20521764). Such N-termini are also acetylated in cells, for instance several Met-Ala-Pro are found to be acetylated by NatF (Aksnes et al., Cell Rep, 2005, PMID:25732826).

We added a corresponding statement to the discussion (line 304-306).

7.- Lines 190-191: 'In both structures, the N-terminal methionine is sandwiched between Leu27 and Tyr145 of Naa30...'

- Lines 313-314: 'In both cases, Met1 is sandwiched in between Tyr145 and Leu27.'

-> Met1 from Gag peptide is stacked between Leu27 and Tyr145, but for Arl3 peptide Leu27 side chain is flipped according to the figures 3c, 3d and 4a. This is not clear from the text and should be clarified, especially as the authors also state that this is an important residue for catalysis.

While we indeed see clear electron density for Leu27 in the Gag peptide, the corresponding density in the Arl3 peptide-containing structure is not well defined, indicating some flexibility. We now show electron density for Leu27 in all structures in Supplementary Fig. 8 and modified our statements accordingly (lines 220-224 and 298-302).

8. - Lines 242-243: 'At peptide position 3, amino acids with a small or elongated side chain are favored, whereas a bulky amino acid appears to reduce binding'.

- Lines 321-322: 'However, a histidine, or, presumably, other bulky amino acids at position 3, hinder catalytic activity.'

-> The lowest K_m reported in this article is for a peptide that has an Arginine at this position. Arginine is certainly bulkier than histidine, so another way to phrase this should be found.

We completely rewrote the corresponding sections in result and discussion (lines 179-206 and 306-314), including the proposed changes.

9. Table 2, Figure 4b.

-> Specificity constant (k_{cat}/K_m) should not be used to compare different enzymes (or variants of the same enzyme), see e.g. PMID: 17433847. Separate plots presenting k_{cat} and K_m would also be more informative.

We now show only k_{cat} for the various mutants in Fig. 4b and report their K_m in the modified Table 2. Furthermore, we labelled Fig. 3a with 'specificity constant'.

Reviewer #2 (Remarks to the Author):

NatC is a key cellular enzyme, which acetylates the N-terminus of cytosolic proteins starting with a Methionine followed by a hydrophobic or amphipathic amino acid, with amino acid in positions 3 and 4 contributing to the selectivity of NatC.

This is a very interesting paper considering the fundamental role of N-terminal protein acetylation by NatC in many aspects of cells and diseases, and the lack of information on its mechanism and structure.

The paper by Grunwald et al., reports several crystal structures of the *Saccharomyces cerevisiae* NatC complex in the apo for CoA-bound form and two different N-terminal peptide-substrates (MFHLV and MLRFV). In the various solved structures, NAA30 adopts the typical GNAT fold, while the small auxiliary subunit NAA38 adopts a Sm fold but without the Lsm-IRG motif that mediates the association with small nuclear RNAs. Even more interestingly, the authors reveal that NAA35 displays a never seen fold, acting as a central assembly hub for NAA30 and NAA38. Strikingly, comparison of NatC with NatA and NatB complexes shows remarkable differences.

Based on ligand-induced structural conformational changes of NAA30 and activity characterization, the authors propose a model of catalytic mechanism for the NatC complex. Finally, the authors identify an electropositive region of NAA35 necessary for the association of NatC with ribosome and propose a NatC-ribosome interaction model.

This work could significantly contribute to the knowledge of Nat enzymes and allow a significant step forward in the field of N-terminal acetylation process, providing mechanistic insights on one of the major Nats.

Thank you very much for the positive feedback!

However, I think that the paper would be strengthened by addressing several issues described below.

Major points

1/ One of my major concerns is related to the NatC-substrate selectivity characterization. Although, the authors have tested several peptides variants, several specific substitutions are missing to fully understand the NatC selectivity. For instance, studies on the effects on the catalytic parameters of a non-favored residue in the position 2, such as Asp/Glu or Arg, in a γ Arl3 (F2D/E/R) or in peptide variants of the protein Gag (L2D/E/R) are missed. This information would certainly clarify the role of residues at position 3, 4 and 5 in the selectivity of NatC. Likewise, no unfavorable substitution at position 4 or 5 has been studied and discussed by the authors, while these specific positions seem to be involved in the formation of the main hydrophobic pocket, important for selecting hydrophobic residues at position 2 in NatC.CoA.MFHLV or NatC.CoA.MLRFV complexes. Likewise, no substitution as F4G with an unfavorable residue at position 4 of the Gag peptide was attempted to confirm the importance of the extended pocket in NatC, as stated by the authors. I suggest that the authors complete their selective characterization, by monitoring the catalytic activity of NatC with additional peptides variants according to what is described above.

Based on this concern, we added new kinetic data for a total of 11 additional peptides to the manuscript (see new Fig. 3a and Table 2). Thus, we included \$\gamma\$ Arl3 peptide variants with substitutions at position 2 (F2K, F2R). Additionally, positions 2, 3, 4 and 5 in both the Gag and \$\gamma\$ Arl3 peptides were systematically exchanged to Glu. Furthermore, a \$\gamma\$ Arl3 variant with the L4A substitution was tested. The corresponding results section was expanded and re-written accordingly (lines 179-206 and 306-314).

2/ In lines 230-240, after comparison with Naa10 of the NatA complexes and the hypothesis that the binding of the substrate has no impact on the catalysis of Naa10, the authors suggest that the NatC.CoA.MLRFV (Gag) structure represents an efficient catalytic conformation of the NatC.. Does this

mean that NatC.CoA.MFHLV (γ Arl3) does not represent a catalytic efficient conformation of NatC, which looks more like an apo conformation?

We discuss this issue in more detail in the revised version (lines 306-314).

The authors analyzed the different amino acid substitutions using the structure of NatC.CoA.MFHLV as reference. Instead, I would suggest analyzing these substitutions using the NatC.CoA.MLRFV.

Due to the very low K_m of the Gag-peptide (<20 μ M), subtle differences in K_m cannot be detected by our assay. Therefore, most peptides were based on γ Arl3, which has a K_m of \sim 140 μ M. Still, we performed new kinetic experiments with 4 additional Gag-peptide variants that harbored glutamate substitutions at positions 2, 3, 4 and 5 (new Fig. 3a and Table 2).

3/ In lines 241-243, the authors claimed that "Based on the presented data, NatC seems to prefer methionine-starting substrate peptides with hydrophobic amino acids at position 2 and 4. At peptide position 3, amino acids with a small or elongated side chain are favored, whereas a bulky amino acid appears to reduce binding". How authors can conclude based on their data that hydrophobic amino acids at position 2 and 4 are favored when no hydrophilic substitution was tried at these positions? In light of the above, the conclusion drawn on the impact of position 3 "At peptide position 3, amino acids with a small or elongated side chain are favored, whereas a bulky amino acid appears to reduce binding" makes no sense to me. The authors should improve this part with further experiments as proposed in point 1 and re-write the text accordingly.

As described above, we tested a variety of new peptides and rewrote the corresponding sections in the results and discussion (lines 179-206 and 306-314).

4/ The authors, using co-sedimentation assay, identified EPR2 Tip/Tip2 as one of the major elements involved in NatC-ribosome bond. Since the presence of NAA35 is necessary to restore NAA30 activity *in vivo*, I suggest the authors to complement the yeast KO NAA35 mutant with the NAA35-EPR2 variant used in the sedimentation assay. This will provide information if EPR2 alone is sufficient to ensure the NatC-ribosome binding.

We performed the requested new experiments and show them in the new Fig. 5d (see also lines 264-268). Initially, we recapitulated the described growth phenotype of NAA35 ko mutant cells on glycerol containing medium at high temperatures (37 °C) and the growth rescue by NAA35 re-expression. Similarly, also the two tip mutants in the EPR2 could rescue the growth phenotype.

Please note that the interpretation of this result is not straightforward, since the molecular basis of reduced yeast growth under non-fermentative conditions is not so clear. It may be envisaged that the EPR mutants do not (fully) disturb the NatC – ribosome interaction *in vivo* or that the NatC-ribosome interaction is not required for the acetylation of all (essential?) substrates, but also other explanations could be imagined. Clearly, more work is required to fully understand the exact role of N-terminal acetylation for the various NatC substrates in yeast and human.

5/ Considering the proposed mechanism for NatC, the authors proposed that Glu29 might be in charge of the second phase of deprotonation p. However, none of their structural data is in favor of this suggestion. Indeed, the authors recognized that the substrate peptide amino group and the carboxyl group of Glu29 are 9 Å away in NatC•CoA•MLRFV and therefore suggested that Glu29, being part of the flexible α 1– α 2 loop, may relocate to the active site. However the authors previously suggested in the manuscript that NatC•CoA•MLRFV was in the efficient catalytic conformation. This is not in accordance with a possible movement of the flexible α 1– α 2 loop. Furthermore, Glu29 is not conserved in different Nats (Fig. 4d), in contrast to the other key residue Y130 (whose key function was first identify by Vetting 2008 in RimI).

In their structures Fig3, Glu29 interacts either with the substrate peptide of residue 3 or with the backbone of residue 4. Since Glu29 could be the involved catalytic base and/or involved in binding the substrate and is far away from its proposed function, it would be better to remove the role of Glu29 in Fig. 4c.

Thanks, we added the Vetting et al. reference (lines 357-359), removed Glu29 from Fig. 4c and toned down the conclusions relating to Glu29 (line 361-369).

Minor points

1/ NAA50 is not active in yeast and recently (Knorr NSMB 2019, Deng 2019, 2020) it was suggested to act as an auxiliary subunit (at least in yeast). Please, clarify this point in the introduction (lines 41-42).

We introduced this suggestion in detail in the discussion (lines 321-326).

2/ Taking into account the recent Arnesen's review, NatC/E/F in human is estimated around 14.4% and 9.6% in yeast. Please, adjust the value (28%) reported in line 50.

The value was taken from an Arnesen review (Ree et al, 2018). A more recent review (Aksnes et al, 2019) states 21% as the estimated size of the human NatC/E/F substrate group, which we now report (line 48).

We were not able to find the suggested values from the referee.

3/ The authors clearly show remarkable differences in the tertiary and quaternary structures of NatC in comparison to those of NatA and NatB. In this context, the authors claimed that $\beta_6\beta_7$ element of the catalytic subunit NAA30, which builds the C-terminal half of the active site of NatC, is not motionless as in NatA or NatB complexes. Indeed, this element is constrained by NAA35 subunit in NatC. Similarly, the remaining N-terminal half of the active site, made of $\alpha_1\alpha_2$ loop, is stabilized differently in NatC than in the NatA or NatB complexes. However, the authors conclude this comparison with the following sentence (line 163-165) "Thus, while the overall architecture of the auxiliary subunits is different in the three NAT complexes, similar mechanistic principles apply for the stabilization of the catalytic center". This last sentence of the paragraph should be changed because it is not in agreement with the whole paragraph and Figure 2. Alternatively, the authors should better explain what are the essential stabilizing elements, similar to those of NatA and NatB, found in NatC catalytic center which allow to suggest a similar mechanistic principle.

As suggested, we removed this statement.

4/ At line 200-2003, the authors wrote "This demonstrates that a methyl side chain at position 2 is, in principle, sufficient to mediate binding. However, such a substrate is unlikely to exist in vivo, as aminopeptidases usually cleave the amino-terminal methionine when it precedes a small amino acid [52]"

I suggest that this part should be re-written taken into account the fact that the specific MAP (Proline at position 3) is a non-optimal MetAP substrate and very often Methionine is not cleaved.

We included this now in the discussion (line 304-306).

5/ Fig. 3b is very difficult to understand. The top view they choose makes it difficult to apprehend the differences between both substrates conformation within the active site and particularly at position 3, 4, 5, highlighted as important for substrate recognition and for which the omit map is less clear. I suggest the authors provide a different view for Fig3b (perhaps side view) and to divide the figure in two parts, one centered on the substrate and the other on the CoA. In addition, a superposition between the two substrates would be useful to understand their binding differences in the active site and values in Table2. As well a superposition between the two CoA in a different view will help to understand the different conformation observed in the structure of both peptide substrates as observed in Fig. 3d.

Based on an overlay of the catalytic subunits, we now show a superposition of the two peptides in the new Fig. 3b. In addition, we modified the electron density representations of the peptides (now

Supplementary Fig. 8) to highlight the different binding modes. Still further, we added a surface representation of the yArl3 peptide to Supplementary Fig. 9.

6/ The section on how NatC interacts with ribosome and the logic used is extremely dangerous and misleading (lines 264-276). The quaternary arrangement of NatC and NatA/E is completely different and currently no information is available on a possible exclusive interaction of these complexes on the ribosomal surface. Therefore, I do not understand the logic to locate NAA30 in an equivalent position of NAA10. It is not surprising to see that the peptide-binding site of NAA30 is therefore found near the ribosomal exit tunnel when NAA30 is aligned on NAA10. Without further results, these models make no sense and this paragraph should be removed from the result section. The authors may discuss this part in the discussion section, but soften their conclusions by stating that additional structural and functional studies are needed to confirm them.

We followed the advice and moved the paragraph to the discussion with a precautionary notion (lines 370-377). The model is based on the idea that the nascent peptide chain may constrain the entry point in the acetylation machinery relative to the ribosomal exit tunnel (see Supplementary Fig. 13c). Furthermore, this model is consistent with the mutagenesis data of EPR2. However, we concur that additional structural and functional studies are required to determine the exact binding mode of NatC to the ribosome.

7/The authors identified Glu118 as the catalytic base with support of Tyr80 but do not discuss Glu29, the replacement of which with Ala or Gln also inhibits NatC activity. Moreover, the equivalent Glu29 residue in NAA10 was previously identified as the catalytic base of NatA. This should be discussed.

We suggested Glu118 as a catalytic base for the first protonation step, based on a superposition with known NAT structures and previous functional studies (lines 342-346). In contrast, Glu29 is positioned far away from the primary amine of the peptide and we therefore did not discuss a potential role for the first deprotonation step. Instead, we expanded the discussion on the role of Glu29 on the stabilization of active sites loops in lines 361-369.

REVIEWERS' COMMENTS

Reviewer #2 (Remarks to the Author):

The authors have now improved their manuscript on the crystal structure and biochemical mechanism of the yeast heterotrimeric NatC complex according to most of my concerns raised in the original review. Yet some minor issues persist and in my view, the authors should consider:

- 1) My major concern remains the physiological role of EPR2 Tip/Tip2. The new complementation experiments clearly demonstrate that EPR2 is not necessary for NatC activity in vivo. In my view, the authors need to seriously convey the experimental limitations and shortcomings of their study related to this point. Since in the rebuttal letter they have proposed several hypotheses of the unnecessary of the EPR2 element in vivo, these should be clearly reported in the result and discussion. So as, I suggest to totally remove from the abstract the sentence relating to the element of binding to the ribosome, since (the authors agree on this) it is necessary to work more to depict its true physiological meaning (if any).
- 2) Please, provide a omit side view of peptides in Supplementary Figure 8a as asked originally (see my point 5).
- 3) Line 193 and 311, please correct the salt-bridge of γ Arl3-His3. This is for me an H-bond.
- 4) Fig. 3, the mutation L4A and consequent increased activity is not discussed, also not compared with the other point mutants (ie., L4F or V5E). Please provide it.

Rebuttal letter

Divergent architecture of the heterotrimeric NatC complex explains N-terminal acetylation of cognate substrates

Grunwald et al.

Thanks for the responses, below you will find our final comments to the concerns of reviewer #2, which are marked in red in the revised manuscript.

REVIEWERS' COMMENTS

Reviewer #2 (Remarks to the Author):

“The authors have now improved their manuscript on the crystal structure and biochemical mechanism of the yeast heterotrimeric NatC complex according to most of my concerns raised in the original review. Yet some minor issues persist and in my view, the authors should consider:”

1) My major concern remains the physiological role of EPR2 Tip/Tip2. The new complementation experiments clearly demonstrate that EPR2 is not necessary for NatC activity *in vivo*. In my view, the authors need to seriously convey the experimental limitations and shortcomings of their study related to this point. Since in the rebuttal letter they have proposed several hypotheses of the unnecessary of the EPR2 element *in vivo*, these should be clearly reported in the result and discussion. So as, I suggest to totally remove from the abstract the sentence relating to the element of binding to the ribosome, since (the authors agree on this) it is necessary to work more to depict its true physiological meaning (if any).

As requested, we expanded the discussion to provide possible explanations for the missing effect of Naa35 tip mutations on the yeast growth phenotype (lines 382-388). We also emphasize the necessity of further experiments for fully understanding the exact role of the co-translational activity of NatC in yeast and human.

However, our NatC-ribosome spin down experiments clearly demonstrate the importance of the electropositive EPR2 region for NatC-ribosome binding *in vitro*. We show in triplicate experiments that two independent NatC mutations in the elongated EPR2 region result in a significant reduction in ribosome association, in contrast to mutations in many other regions. Thus, we believe that our claim “we [...] identify a ribosome-binding patch in an elongated tip region of NatC” is well supported by our *in vitro* experiments and structural data.

2) Please, provide a omit side view of peptides in Supplementary Figure 8a as asked originally (see my point 5).

We added the requested 90°-rotated side view of the omit map in the modified Supplementary Fig. 8a.

3) Line 193 and 311, please correct the salt-bridge of γ Arl3-His3. This is for me an H-bond.

We now refer to the mentioned bond as “hydrogen bond” in lines 194 and 314.

4) Fig. 3, the mutation L4A and consequent increased activity is not discussed, also not compared with the other point mutants (ie., L4F or V5E). Please provide it.

We now describe the L4A and L4F mutants in both the results and/or discussion (lines 203, 316, 318). As the effects of the V5E mutation was only moderate, as mentioned in the results (line 208), we did not further elaborate on it in the discussion.